# Explaining heterogeneity in medial entorhinal cortex with task-driven neural networks

**Aran Nayebi**[1,*], **Alexander Attinger**[2], **Malcolm G. Campbell**[2], **Kiah Hardcastle**[2], **Isabel I.C. Low**[1,2,7], **Caitlin S. Mallory**[2], **Gabriel C. Mel**[1], **Ben Sorscher**[4], **Alex H. Williams**[6,7], **Surya Ganguli**[4,7,8], **Lisa M. Giocomo**[2,7], and **Daniel L.K. Yamins**[3,5,7]

[1]Neurosciences Ph.D. Program, Stanford University
[2]Department of Neurobiology, Stanford University
[3]Department of Computer Science, Stanford University
[4]Department of Applied Physics, Stanford University
[5]Department of Psychology, Stanford University
[6]Department of Statistics, Stanford University
[7]Wu Tsai Neurosciences Institute, Stanford University
[8]Facebook AI Research, Facebook, Inc.
[*]Correspondence: `anayebi@stanford.edu`

## Abstract

Medial entorhinal cortex (MEC) supports a wide range of navigational and memory related behaviors. Well-known experimental results have revealed specialized cell types in MEC — e.g. grid, border, and head-direction cells — whose highly stereotypical response profiles are suggestive of the role they might play in supporting MEC functionality. However, the majority of MEC neurons do not exhibit stereotypical firing patterns. How should the response profiles of these more "heterogeneous" cells be described, and how do they contribute to behavior? In this work, we took a computational approach to addressing these questions. We first performed a statistical analysis that shows that heterogeneous MEC cells are just as reliable in their response patterns as the more stereotypical cell types, suggesting that they have a coherent functional role. Next, we evaluated a spectrum of candidate models in terms of their ability to describe the response profiles of both stereotypical and heterogeneous MEC cells. We found that recently developed task-optimized neural network models are substantially better than traditional grid cell-centric models at matching most MEC neuronal response profiles — including those of grid cells themselves — despite not being explicitly trained for this purpose. Specific choices of network architecture (such as gated nonlinearities and an explicit intermediate place cell representation) have an important effect on the ability of the model to generalize to novel scenarios, with the best of these models closely approaching the noise ceiling of the data itself. We then performed *in silico* experiments on this model to address questions involving the relative functional relevance of various cell types, finding that heterogeneous cells are likely to be just as involved in downstream functional outcomes (such as path integration) as grid and border cells. Finally, inspired by recent data showing that, going beyond their spatial response selectivity, MEC cells are also responsive to non-spatial rewards, we introduce a new MEC model that performs reward-modulated path integration. We find that this unified model matches neural recordings across all variable-reward conditions. Taken together, our results point toward a conceptually principled goal-driven modeling approach for moving future experimental and computational efforts beyond overly-simplistic single-cell stereotypes.

35th Conference on Neural Information Processing Systems (NeurIPS 2021).

# 1    Introduction

From exploring new areas, planning shortcuts, and returning to remembered locations, the ability to self-localize within an environment subserves a range of navigational behaviors that are essential for survival. The hippocampus (HPC) and medial entorhinal cortex (MEC) are known to contain cells that encode the position of an animal by displaying firing fields with strikingly regular response patterns, influenced by self-motion [O'Keefe and Dostrovsky, 1971, Hafting et al., 2005, Kropff et al., 2015, Solstad et al., 2008, Sargolini et al., 2006]. For example, MEC grid cells possess characteristically symmetric and periodic tuning curves, yielding hexagonal arrays of neural activity over physical space. Additionally, border cells that fire maximally near environmental boundaries and head direction cells that fire only when an animal faces a particular direction are among the other MEC cell types with interpretable tuning curves.

However, a large fraction of the MEC population have unconventional and heterogeneous tuning to navigational variables and are less obviously well-described in terms of simple, stereotypical tuning patterns [Hinman et al., 2016, Hardcastle et al., 2017]. How can we characterize what these other, more heterogeneous, populations of cells do? Are they critical to the abilities of the animal in real-world memory tasks? And if so, how is their role distinct from the more stereotypical grid-like cells? Here, we take a quantitative computational modeling approach to answering these questions.

To start with, we address the question of whether there is a phenomenon to model in the first place, and how one would measure model accuracy quantitatively. To this end, we identify the *similarity transform* between neural populations in different animals — that is, a mapping which takes neuronal population vectors in a "source" animal and maps it to corresponding neuronal population vectors in a "target" animal. To the extent that robust similarity transforms can be found that match up MEC population responses across multiple animals measured in multiple experimental conditions, then a reliable pattern of neural response behaviors (across conditions) has been identified. This identification strategy is well-defined even when there is no known *a priori* taxonomy of functional response types. We find here that with the right mapping class (mostly Ridge-like linear regression), the inter-animal consistency of MEC neuronal responses is in absolute terms very high ($> 0.8$), and in relative terms just as high as for heterogeneous cells as for more stereotypical grid cells.

Using this same similarity transform, we then evaluate the ability of each of multiple computational models to explain response variance of MEC neurons, treating each candidate model as a potential "source animal" and measuring how well it maps to each target real animal. We look to the recent literature to identify potential candidate models. Over the past several decades, collaborations between experimental and computational neuroscientists have led to the formulation of dynamical-systems models of grid cell formation, and helped illustrate possible functional roles for grid cells in supporting path integration-based hippocampal place cells [Skaggs et al., 1992, Zhang, 1996, Fuhs and Touretzky, 2006, Burak and Fiete, 2009]. These powerful models make a number of non-obvious predictions about MEC neural properties, some of which have been confirmed in subsequent experimental work [Ocko et al., 2018, Campbell et al., 2018]. Despite their success, such models are limited in their explanatory scope, hand-designed to capture the properties of one stereotypical cell-type class (e.g. grid or border cells) at a time, or combinations of several cell types via multiple dedicated type-specific modules [Couey et al., 2013, Yoon et al., 2013].

A potential solution to this problem arises out of recent work creating learned neural networks that achieve path integration [Cueva and Wei, 2018, Banino et al., 2018]. Intriguingly, these models have been found to contain internal units that resemble grid cells, suggesting that such stereotypical cells embody a computational solution to path integration that naturally arises from satisfying an end-to-end functional constraint. Recent work has demonstrated that the underlying mathematical reason for this fact is due to pattern forming dynamics under a nonnegativity constraint [Sorscher et al., 2019, 2020]. Intriguingly, in addition to having units that resemble stereotypical grid or border cells, these learned neural networks also naturally possess a wide variety of other less easily described unit types, raising the possibility that these artificial "heterogeneous" units might be somehow resemble the actual heterogeneous cells making up the majority of real MEC populations.

Motivated by these ideas, we generate a wide variety of candidate model networks by varying architectural structure and end-to-end optimization objectives, each expressing a different hypothesis for MEC circuit structure and function. Architecturally, we formulate variants based on using different types of nonlinearities and different types of local recurrent memory circuits (e.g. RNNs [Elman,

1990], UGRNNs [Collins et al., 2017], GRUs [Cho et al., 2014], and LSTMs [Hochreiter and Schmidhuber, 1997]). From a task point of view, we test both simple dimensionality reduction [Stachenfeld et al., 2014, Dordek et al., 2016] as well as place cell mediated vector path-integration [Banino et al., 2018] and direct position estimation [Cueva and Wei, 2018].

Our core result is that there is substantial variation in the models' abilities to explain MEC responses, especially those of the heterogeneous cells, as a function of a model architecture and task. Some models, such as the classic "Grid Cell" model based on low-rank decomposition of place cell fields, do a reasonable job explaining grid cell responses but are quite poor at explaining most other neurons. Task-optimized learned models typically do better, especially those optimized for place cell-mediated vector path-integration. The best model – with memory-gated rectified nonlinearities – essentially *solves* the neurons, capturing nearly 100% of the noise ceiling of the data. This is of substantial interest, given that the nonlinear components of this model are not directly optimized to match neural data (just to solve the task), and given that a series of strong control models capture much less of the MEC neural variability. This same model also best generalizes to a variety of novel experimental conditions, and has the best match to the empirical data on a grid score distribution metric.

With this predictive model in hand, we then begin to address our second core question: what is the functional role of heterogeneous neurons? We generate several results suggesting that heterogeneous neurons are important for path integration, including cell-type specific virtual knockout experiments, in which we compare performance degradation when deleting grid and/or border cells as compared to heterogeneous units. Overall, we find that models are quite robust to knockouts, and differences between heterogeneous and stereotypical cell knockouts are very small, suggesting that stereotypical cell-types may not be especially more functionally important than units with less easily-characterized response profiles.

Building on the above results, we extend models to encompass MEC cell responses as a function of reward as well as spatial position, introducing a simple modeling paradigm that performs reward-modulated foraging in the context of the path-integration task. We find that this unified task-optimized model matches neural responses across all reward and spatial conditions, and that a reward-response mechanism at an intermediate point on the explore-exploit continuum best explains neural responses. Taken together, our results suggest how specific processes of biological performance optimization may have directly shaped the neural mechanisms in MEC as a whole, and provides a path for enlarging the study of MEC beyond overly-restrictive response stereotypes.

## 2 Reliability of heterogeneous cell response profiles

What firing patterns of MEC cell populations are common across multiple animals, and thus worthy of computational explanation? This question is comparatively straightforward for stereotypical MEC cells, because the very presence of these stereotypical features (e.g. hexagonal grids of a given orientation and spatial frequency) allows the definition and measurement of observables that arise reliably across trials and animals (e.g. the grid score distribution). But how can this be done generally for populations in which one does not have a prior characterization of what each cell encodes? We take inspiration from methods that have proven useful in modeling visual, auditory, and motor cortex [Yamins and DiCarlo, 2016, Kell et al., 2018, Michaels et al., 2020]. Specifically, we aim to identify the *narrowest* class of similarity transforms needed to map the firing patterns of one animal's MEC population to that of another (Fig. 1a). As with other cortical areas, this transform class likely cannot be so strict as to require fixed neuron-to-neuron mappings between MEC cells, since even within the same animal at different times, MEC and HPC populations can undergo remapping that shift cell responses across the population [Farhoodi et al., 2020, Low et al., 2020]. However, the transform class for MEC also cannot be so loose as to allow a completely unconstrained linear mapping, since the highly structured response patterns of stereotypical MEC cell types such as grid cells may not be guaranteed to be preserved under arbitrary linear transforms.

To identify this transform class, we utilize data collected from electrophysiology in 12 awake behaving mice ($n = 620$ cells) performing open foraging for randomly scattered crushed cereal in a $100cm^2$ 2D arena [Mallory et al., 2021]. We explore a variety of mapping transform classes between the population rate maps of these neurons (Fig. 1b). As a baseline we evaluate a strict one-to-one mapping transform, in which each target unit is mapped to the single most correlated unit in the source animal. We also evaluate more powerful linear transforms, including Lasso, Ridge, and ElasticNet regression,

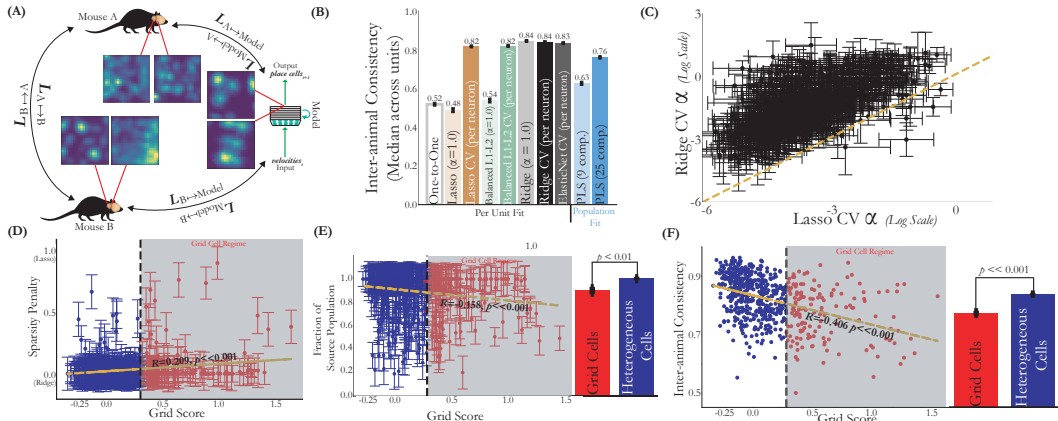

Figure 1: **Heterogeneous cells are just as reliable as grid cells across 2D environments and task conditions. (A)** We establish inter-animal consistency levels by mapping animals to each, using inter-animal transforms $L$ of at-most-linear functional complexity. Computational models are then mapped to real animal's MEC data using the same transform class. **(B)** Inter-animal consistency levels as assessed with mapping transform classes of different complexity. Median and s.e.m. across 620 cells. **(C)** The alpha value per neuron as chosen by Ridge CV plotted against the alpha value chosen by Lasso CV, on a log scale. Median and s.e.m. across ten train-test splits. The unity line is in yellow. **(D)** Per unit sparsity penalty under ElasticNet CV (grey bar in (B)) as a function of grid score. Median and s.e.m. across ten train-test splits. **(E)** The fraction of source units assigned nonzero weight per target cell plotted against its grid score (median and s.e.m. across ten train-test splits). Red denotes grid cells with grid score $> 0.3$ and blue denotes heterogeneous cells with grid score $\leq 0.3$. The bar plot on the right is the mean and s.e.m. of this median quantity for the identified grid and heterogeneous cell populations. The $p$-value is obtained from an independent $t$-test across neurons between populations. **(F)** Inter-animal consistency of each unit plotted against its grid score. The bar plot on the right is the mean and s.e.m. of this quantity for the identified grid and heterogeneous cell populations. The $p$-value is obtained from an independent $t$-test across neurons between populations.

as well as population-level Partial Least Squares (PLS) regression. For all methods with fittable parameters, mapping fit is performed on a random 20% of spatial position bins, and evaluated on the remaining 80% of position bins (see supplement for more details).

The strict one-to-one mapping yielded low inter-animal consistency among the maps considered, capturing around 50% of the target neural response variability. Linear regression with strong sparseness priors, such as Lasso (L1 penalty) and balanced Lasso-Ridge (equal L1 and L2 penalty) regression, also proved to be too strict when evaluated with fixed regularization level $\alpha = 1$, yielding hardly any improvement over the one-to-one mapping. However, pure Ridge (L2 penalty) regression at this regularization level was highly effective, recovering nearly all target variability for most neurons. Cross-validating the L1 regularization constant on a per-cell basis improved the fits, at the cost of requiring substantially looser regularization than for L2 (Fig. 1c). Under an ElasticNet mapping in which both sparsity penalty and regularization constants were chosen with cross validation, most cells were generally still best explained by an essentially Ridge-like transform with no sparsity penalty (Fig. 1d). However, different target cells required different numbers of source units to achieve effective mapping (Fig. 1e). As expected, cells with more stereotypically grid-like response patterns on average chose a higher sparsity penalty (slight positive slope in Fig. 1d) and required fewer source cells to capture (slight negative slope in Fig. 1e) as compared to heterogeneous cells, though this effect is weak. Critically, heterogeneous cells did not have lower inter-animal consistency than grid cells (Fig. 1f).

These results show that the heterogeneous non-stereotyped cell populations are reasonably similar across animals – at least up to (mostly Ridge-regularized) linear transform – establishing that there is a reliable target pattern to study in the first place.

## 3 Task-optimized models of MEC spatial response variation

**Evaluating a spectrum of candidate models.** We evaluated models of several basic types. First, we considered learnable neural networks, all of which accept a stream of two-dimensional velocity input vectors, have a single layer of hidden neurons identified as the putative MEC population, and

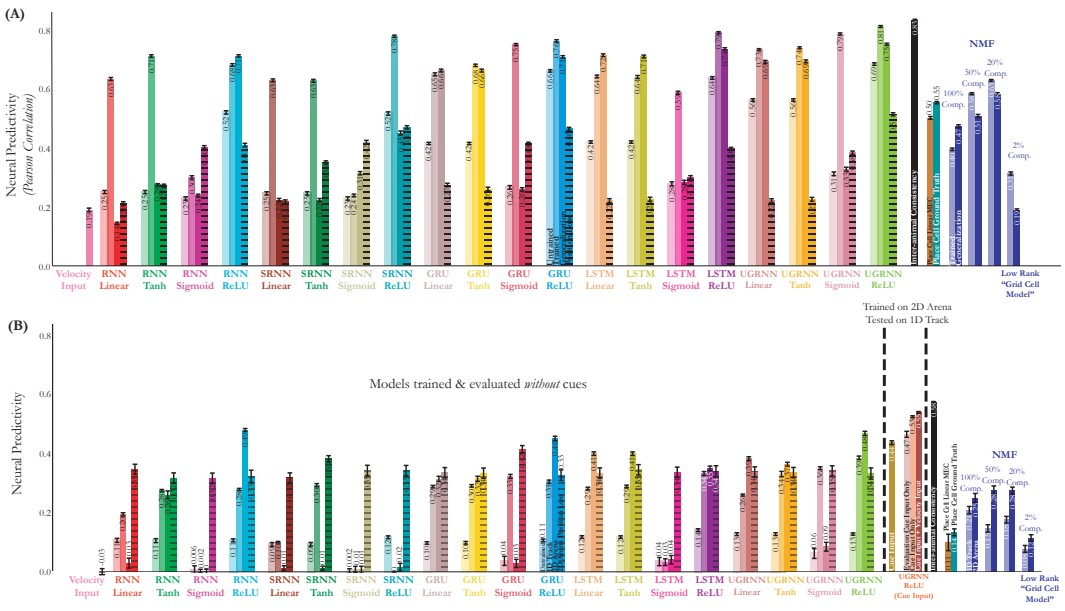

Figure 2: **Task-optimized navigational models best predict the entire MEC population. (A)** Neural predictivity of the model MEC units to the real MEC responses of 12 animals in a $100cm^2$ 2D open field under the ElasticNet CV regression transform (grey bar in Fig. 1b). Median and s.e.m. across 620 total units. **(B)** Same as (A), but now the models are evaluated against responses of MEC units while the animal is traversing a $400cm$ 1D track. "1D Track" refers to models trained to path integrate on this same 1D track. "2D Arena" refers to models trained to path integrate on the $2.2m^2$ arena and evaluated on the 1D track. Median and s.e.m. across 2861 units from 8 different animals.

which are optimized to perform some form of path integration readout on simulated motion paths (Erdem and Hasselmo [2012]) in a fixed-sized arena. The networks varied according to the nature of their local recurrent cell structure (ungated RNN, UGRNN, GRU, or LSTM), activation functions (Linear, Tanh, Sigmoid, or ReLU), and output objective function (explicitly constructing a set of simulated place field neurons [Banino et al., 2018] or directly performing two-dimensional position integration [Cueva and Wei, 2018]). Following Banino et al. [2018], all networks consisted of three nonlinear layers, though for the ungated RNN we included its original two layer version (following Cueva and Wei [2018]) and a three layer version ("SRNN"). We ensured that comparisons were fair by equalizing the size of the hidden layer across models, and included an architecture-only control with untrained filter weights.

We also implemented a class of models describing MEC activations as Non-negative Matrix Factorization (NMF) operating on simulated place cells. The lowest-rank version acts a "Grid Cell" model, inspired by recent work positing MEC as a low-dimensional embedding of hippocampal place fields [Stachenfeld et al., 2014, Dordek et al., 2016], and further validated by the mathematical theory developed in Sorscher et al. [2019, 2020] to explain the emergence of grid cells. Higher-rank NMF models titrate between the low-rank Grid Cell model and a full-rank control that measures how well MEC cells can be explained as a linear projection of their putative place cell outputs.

For evaluation, we map each model's proposed MEC activations to empirically measured firing patterns of the real neurons, using the same type of cross-validated ElasticNet regression transform used to measure inter-animal similarity as in Section 2. The predicted neural responses under the mapping are then compared to real target neural responses on a neuron-by-neuron basis, and the median of accuracy of these predictions taken over target neurons. In addition to evaluating model-data match when constructed on arenas of the same size in which the neural data were collected ($100cm^2$), we also evaluated models trained on larger arenas ($2.2m^2$) but tested on the $100cm^2$ arena.

**Neural prediction results.** The results of our model evaluation, shown in Fig. 2a, support several inferences:

- Different models are substantially different in their ability to predict neural responses. MEC electrophysiology data collected during 2D open-field foraging is thus a strong model target that effectively separates candidate models from each other.

- Task-optimized models are reliably better than their untrained controls, across all architecture types and objective functions.
- Rectification is substantially better than Linear, Tanh, or Sigmoid activation, especially for promoting generalization to new arena sizes.
- Under the rectification nonlinearity, gated circuit architectures (UGRNN, GRU, LSTM) improve model fits compared to the simple ungated alternatives (RNN and SRNN).
- The explicit place cell construction task leads to substantially better fits than the direct two-dimensional path integration task ("Position Loss"), even though both were reported in the literature to create grid cell-like units. Enforcing a place field representation at the output of the network thus appears to be an important constraint in order to recapitulate responses in MEC.
- The Grid Cell (low-rank NMF) model is a poor fit to the MEC population overall.
- Full-rank NMF and "Place Cell Linear MEC" output-based controls capture approximately half the explainable variance of MEC neurons, significantly more than the simple velocity linear input control, but substantially less than any of the input-driven ReLU networks.
- The best model is the UGRNN ReLU trained with the place cell loss ("UGRNN-ReLU-Place Cell"). This model captures nearly 100% of the explainable variance of the MEC neural population response when trained for place cell construction on the arenas of the same size as that on which neural data was collected.

**Generalization to novel experimental conditions.** If a model is truly correct, then once trained, it should capture neural responses in any new tested condition. We tested generalization in two ways. First, as shown in the second from the right bars of Fig. 2a for each model architecture class, we performed a 2D arena size generalization test by constructing models on one arena size, testing against neural data on another. We found this generalization test gives essentially the same rank-order comparison as in the same-arena-size test, with one striking exception: the RNN Tanh model performs well within arena size, but fails to generalize. The UGRNN-ReLU-Place Cell model again performs the best, capturing 90% explained variance of the neural responses on the novel arena size.

Second, we also evaluated the same 2D-pretrained models by running them on a 1D track, comparing models to neural data collected from mice in a 1D virtual reality setup, using the same ElasticNet mapping procedure as in the 2D comparisons (see supplement for more experimental details). We found (Fig. 2b) that 2D arena trained model results had some similar rank order as for the original 2D results (0.43 Spearman rank correlation), with the UGRNN-ReLU-Place Cell model trained in the 2D arena (and then evaluated on the 1D track) achieving the best match (82%) when trained for place-field construction. We hypothesized that the larger gap between the best model and the inter-animal noise ceiling, as compared to the 2D case, was due to the fact that during the 1D experimental data collection, mice were also presented with visual cues. MEC neurons are known to respond to visual input, but as none of our evaluated models had a visual front-end, they could not respond accordingly. To test this hypothesis we trained the UGRNN-ReLU-Place Cell model in 2D, but with phantom visual cue locations added as input while performing the path integration task. We then evaluated this model against 1D neural prediction (with the real visual cue locations as input), and found that including cues rescued model performance back to essentially the noise ceiling, as in 2D (Fig. 2b, orange bars).

**Assessing grid and border score distribution match.** The *grid score* is a metric of how stereo-typically grid-like the response pattern of a given unit is (see supplementary material for specific definitions), with "grid cell" typically defined as having a grid score of greater than 0.3. Similarly, the border score is a measure of how responsive a given unit is in the presence of environmental boundaries, with a "border cell" typically defined as having a border score greater than 0.5 [Solstad et al., 2008]. The ground-truth distribution of grid and border scores for cells in an unbiased experimental population (gray bars in each subpanel of Fig. 3a,c) characterize the extent to which real MEC populations are non-stereotypical in their responses. To further assess model accuracy, we also compared the distribution of grid and border scores within each model to that of the real data, using the (negative) Kolmogorov-Smirnov (KS) distance of the empirical and model grid score distributions as a quantitative metric. This metric is both stronger and weaker than the mapping accuracy metric used above — stronger in the sense that, since there is no parameter fitting in the metric, to fit it well a model has to have the correct distribution in its raw feature output; and weaker in that it only assesses cells on one component of their profile (i.e. "grid-ness" or "border-ness"). We observed that the low-rank Grid Cell model has poor fit on this metric, essentially because it contains *too many* grid cells (see Fig. 3a, upper left). In contrast, the same model that achieves best fits on the neural-fit

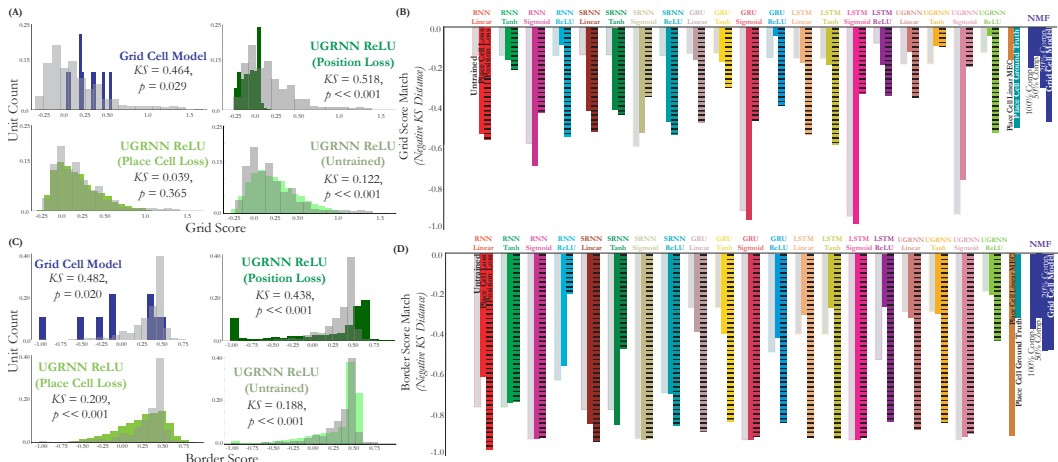

Figure 3: **Relationships to grid and border score distribution. (A,C)** Example distributions of grid and border scores from selected models against the ground truth distribution in the neural data (grey). Negative Kolmogorov-Smirnov distance between the distribution of grid scores of the model units to those of the 620 units in the data. The models have all been trained in a different $2.2m^2$ arena, but are evaluated on a $100cm^2$ environment that the animals are in. **(B, D)** Quantification across all models.

metric (UGRNN-ReLU-Place Cell) also achieves the best on the grid cell distribution match metric, and is the only architecture that cannot be distinguished from the ground truth at the $KS < 0.05$ significance level. Across model architectures, KS-distance was generally in line with neural regression fit metrics (see Fig. 3b,d and Fig. S2). One key difference between the metrics, however, is that for most architectures, the untrained filters provided *better* grid score distribution matches as models with task-trained parameters — except for the that best matched architecture, UGRNN ReLU, where the trained place cell construction model is better than its untrained counterpart (and similar to it for the border score) — and models trained for direct path integration were especially poor on this metric. These results suggest the two metrics in Figs. 2 and 3 are complementary which aspects of model correctness they address, and together help zero in on the most effective models overall.

## 4 Predicting the functional relevance of heterogeneous cells

**Correlating task performance and neural predictivity.** To begin to address the question of the functional relevance of heterogeneous cells, we first looked at the overall correlation between model task performance and neural fit (Fig. 4a). Though the correlation is imperfect, the most task performant models (e.g. UGRNN- and LSTM-ReLU-Place Cell) achieve the best matches to neural fit suggesting that improved task performance may be causally related to the ability capture neuron response patterns across the population. In contrast, there is a comparatively weaker relationship between grid and border score distribution match and model performance (Fig. 4b). While the most task performant models achieve the best match here, models with a gating architecture and ReLU are strong matches to these metrics with untrained filters, illustrating the importance in matching cell properties other than "grid-ness" or "border-ness" in predicting task performance.

**Relative predictivity gain for heterogeneous cells.** The above result is put into greater perspective by comparing neural predictivity differential between a task-trained UGRNN-ReLU-Place Cell model and the Grid Cell model, on a per-neuron basis, as a function of grid score (Fig. 4c). The task-trained model has improved neural predictivity relative to the Grid Cell model for grid cells (grid score $> 0.3$) and heterogeneous cells, but the improvements on the latter (as well as border cells, Fig. S3) are larger than for grid cells. This again suggests that the heterogeneous cells are playing a substantial role in allowing the trained model to achieve improved performance.

**Virtual knockout experiment.** To address the question most directly, we performed a cell-type-targeted virtual knockout comparison experiment. In doing this, we used the UGRNN-ReLU-Place Cell and LSTM-ReLU-Place Cell models, the two best models emerging from the previous section with essentially similar neural predictivity across multiple metrics. We identified the units in the trained model with high grid score ($> 0.3$) or border score ($> 0.5$), and gradually increased the threshold, while measuring task generalization performance (see supplement for details of this

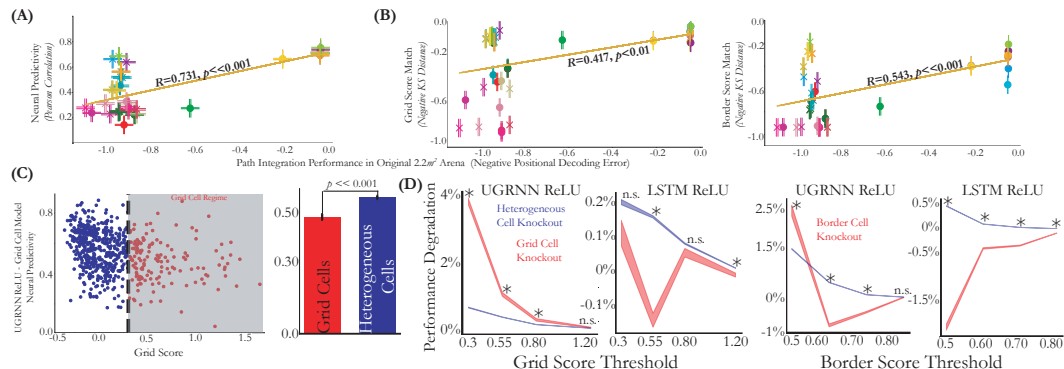

Figure 4: **Heterogeneous cells are relevant to navigation.** **(A)** Neural predictivity (median and s.e.m. across 620 units) versus path integration performance (mean and s.e.m. across 20,000 episodes), measured by negative positional decoding error, of the neural network models. The models are either untrained ("X") or trained with the place cell loss ("O") in the $2.2m^2$ arena. Positional decoding error is measured by taking the top 3 most active place cell outputs in each model, evaluated on the $100cm^2$ arena. **(B)** Same as (A), but with grid score (*Left*) and border score (*Right*) distribution instead of neural predictivity. **(C)** (*Left*) Per unit neural predictivity difference between the UGRNN-ReLU-Place Cell and Grid Cell models trained on the $2.2m^2$ arena and evaluated on the $100cm^2$ arena, plotted against that unit's grid score. (*Right*) Quantification of this difference aggregated across the grid cell and heterogeneous cell populations, respectively (mean and s.e.m.). The $p$-value is obtained from an independent $t$-test across neurons between populations. **(D)** For the UGRNN-ReLU-Place Cell and LSTM-ReLU-Place Cell models, we measure the normalized performance degradation as evaluated on the $100cm^2$ arena, relative to the full model trained on the $2.2m^2$ arena with the place cell loss. We identify units in this trained model with grid and border scores of varying thresholds and knockout the same number of heterogeneous cells (randomly sampled 100 times), to yield the "heterogeneous cell knockout" (blue). The $x$-axis denotes the threshold used for the score. Mean and s.e.m. over evaluation episodes. * denotes a $p$-value $< 0.01$ obtained from an independent $t$-test between the performance degradations of the two knockouts at a given grid or border score threshold.

knockout procedure). We similarly ablated matched numbers of heterogeneous units (grid score $\leq 0.3$ or border score $\leq 0.5$ or both, see Fig. S4 for the latter), again measuring model performance.

The main result of this experiment (Fig. 4d) is that all networks are highly robust to knockouts, experiencing only at most 1-4% performance degradation relative to the full model even when substantial fractions of units are knocked out (corresponding to 21-24% of a single layer's units for grid score $> 0.3$ and 16-20% of these units for border score $> 0.5$). At stricter grid and border cell thresholds, the heterogeneous knockout is similarly injurious to the cell-type specific knockouts, across the two model architectures, suggesting that highly-stereotypical "classical" border and grid score cells are not more essential to the path integration behavior than heterogeneous cells. At low thresholds (when counting many relatively heterogeneous cells as "grid" or "border" cells), the two model architectures give divergent predictions, with the UGRNN model showing a small but significant effect of grid cells relative heterogeneous cells, and the LSTM model showing the opposite. It would be of substantial future interest to confirm or reject either of these models' predictions with a real targeted knockout experiment *in vivo*.

## 5   Modeling reward-driven modulation in MEC

Recent work in both rodents and humans has uncovered that MEC and HPC neurons represent not only literal space, but also capture spatialized layouts in a more abstract sense in modalities other than spatial position [Constantinescu et al., 2016, Aronov et al., 2017]. It has also been seen [Butler et al., 2019, Boccara et al., 2019] that non-spatial rewards can influence the shape of MEC response profiles, restructuring them to incorporate the location of the learned reward. These intriguing phenomena represent a natural direction for modeling, but are not captured by any of the neural network models described in Section 3, as they do not have reward-modulated inputs. (Note that while Banino et al. [2018] used a pretrained LSTM-Tanh-Place Cell network as a front-end on which reinforcement-based navigation tasks are evaluated downstream, reward state is not input to their network or otherwise propagated back into the MEC-like layers of their model, and thus cannot address the neural modeling question raised here.)

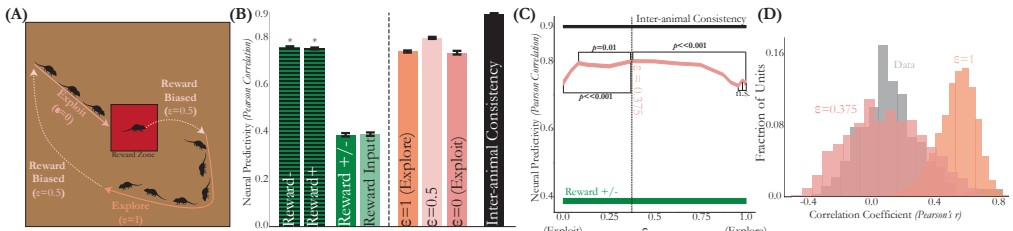

Figure 5: **Reward biased path integration captures remapping of responses in the presence of a reward.** **(A)** Schematic of the three conditions. "Explore" is the original set of random bout trajectories used to train the agent as before. "Exploit" involves the agent navigating directly to the reward zone (red box) within a fixed number of timesteps per episode (7), and spending the remaining 13 timesteps only path integrating in the reward zone. "Reward Biased" refers to training with each of the above two trajectories on $0 < \varepsilon < 1$ of the training episodes. **(B)** Neural predictivity of the UGRNN-ReLU-Place Cell model, evaluated on the $150cm^2$ arena. Median and s.e.m. of 598 cells from 7 rats. Green bars denote the model evaluated with random walk velocity inputs. $*$ denotes comparison of the model to each condition *separately*. Coral bars (to the right of the vertical line) denote the model evaluated with random walk and reward biased velocity input modulation. **(C)** Neural predictivity of the reward biased UGRNN-ReLU-Place Cell model as a function of $\varepsilon$. Median and s.e.m. of 598 cells from 7 rats. The $p$-value is obtained from an independent $t$-test across neurons between selected model pairs in brackets. **(D)** Histogram of the correlation coefficient of unit rate maps between the random foraging and velocity-input modulated reward condition in the $\varepsilon = 0.375$ and $\varepsilon = 1$ UGRNN-ReLU-Place Cell models. The same metric applied to the data is shown in grey.

We thus sought to build new network models that respond to the existence of an extrinsic reward in a behaviorally-meaningful fashion. One natural approach would be to build a full reinforcement-learning (RL) agent in which reward depends on environmental features (e.g.total number of food units encountered during a forage path). A model optimized end-to-end to forage under these conditions could be successful in creating a representation that matches MEC reward-response modulation. However, we took a more direct approach: simply change the behavior of the agent in response to reward in the way RL would be expected to do if successful, by modifying the foraging trajectories used during training and testing to no longer be pure random walks.

Specifically, we trained the UGRNN-ReLU-Place Cell model with a variety of foraging policies titrating between pure exploration and exploitation. Paths could either purely exploit a known reward resource by directly moving to the reward location when it is present (Fig. 5a, $\varepsilon = 0$ condition), purely explore with a random walk as in the original unmodulated model ($\varepsilon = 1$), or take an intermediate policy ($0 < \varepsilon < 1$). In creating these scenarios, we sought to roughly mimic experimental observations showing animals often take rapid and direct paths to the reward zone [Butler et al., 2019] (see supplement for details of implementation). Throughout, the actual training task of the network remained the same as above — place cell construction — but the network now would have to be tolerant to reward-modulated input velocity changes while maintaining positional knowledge.

We then compared these networks to neural data from [Butler et al., 2019] for animals in conditions both with reward (**reward+**) and without it (**reward-**), in which the animal navigates to a $20cm^2$ reward zone within a $150cm^2$ arena to receive 0.5-1 units of cereal. We first performed inter-animal consistency checks, using the same method as we did with the purely spatial-condition data in Section 2 above, finding that inter-animal consistency for neural responses is high both within reward condition (e.g. just spatial modulation in **reward+** and **reward-** separately), and across both spatial and reward-modulated conditions overall (Fig. S6).

As a baseline, we then evaluated the ability of the original (non-reward-modulated) UGRNN-ReLU-Place Cell model to match neural data. Consistent with results reported in the previous sections, this model achieved high neural predictivity in **reward+** and **reward-** separately (Fig. 5b, hatched bars). However, as expected, this non-modulated model was ineffective at predicting response patterns across both reward conditions (Fig. 5b, **reward+/-**). Moreover, simply augmenting the network input to receive an additional binary reward state (Fig. 5b,"Reward Input"), but without using that input to specifically modulate input velocities or output behavior, did not substantially improve neural predictivity, showing the need for nontrivial integration of reward-modulated state.

We then evaluated the reward-modulated networks, generating network outputs for comparison with the **reward+** and **reward-** conditions by modulating velocity inputs to match each condition during testing (see supplement for details). We found that the pure-explore ($\varepsilon = 1$) model is substantially

better at matching neural responses across reward conditions than the **reward+/-** baseline. This is perhaps somewhat surprising since in this comparison the underlying neural network model is identical, just evaluated with or without reward-modulated input data during testing. This result suggests that a substantial fraction of the reported reward-modulated effect in MEC may actually simply be input-driven, lending a new interpretation to results of [Butler et al., 2019]. (It may be useful to note that this result represents a model-based control that was inaccessible to the authors of [Butler et al., 2019].) However, we did find that exposing the network to a mixture of exploration and exploitation behaviors during training does lead to networks with somewhat improved neural predictivity. Results were largely robust to the specific proportion of exploration-vs-exploitation (Fig. 5c), though the best model (at $\varepsilon = 0.375$) model was statistically-significantly better than alternatives and had a substantially closer match ($KS = 0.13$ vs. $KS = 0.81$) to the data's unit-level remapping across conditions than the original $\varepsilon = 1$ model (Fig. 5d). These results are consistent with there being some nontrivial within-MEC reward-modulated responses beyond simple input modulation alone.

## 6 Discussion

We have identified a goal-driven neural network model of MEC that is quantitatively accurate across a wide variety of common experimental conditions, and that can be used to generate nontrivial insights about the underlying mechanisms and functional roles of mouse MEC. More generally, our results suggest that constraint-driven neural networks may provide a fruitful approach for studying navigation and memory in the MEC, and beyond.

Our work suggests the existence not of a specialized class of heterogeneous cells that is functionally segregated from classic cell types, but rather a continuum of cells within a single unified network that naturally encompasses grid, border, and heterogeneous cells. Future research on the computational foundations of MEC/HPC may thus be well-served by putting less emphasis on identifying cleanly-stereotypical cell types (such as grid cells) or perfecting mathematically simple models of single such cell-types, and looking instead for holistic computational ideas that move beyond "easy to visualize" but perhaps overly-simplistic tuning-curve categories.

The nature of the understanding afforded by such a modeling approach comes from their ability to make inferences about what constraints (both structural and functional) are consistent with the data. The current work rules out a variety of simple network connectivity diagrams as inconsistent with MEC cell data, and narrows the space of functional goals MEC circuit weights might be optimized for over evolutionary timescales. Improving the results here will hopefully narrow these constraints yet further, with the ultimate goal of identifying constraints that yield the uniquely correct MEC circuit diagram and synaptic weights – or at least, the narrowest set of such networks consistent with the inherent variability between animals in the real population.

There are some key limitations, however, on our results. First, we attempted to identify the simplest underlying transform that would map animals within a population to each other, using this as the basis for conclusions both about unit-type reliability and model-data comparisons. While the philosophy of this approach may be sound [Cao and Yamins, 2021a], in practice it is possible that we could narrow the transform class further by (e.g.) enforcing that it be invariant to one more more properties of classical cell-types (e.g. periodicity). Finding algorithms to better identify sharp inter-animal transform classes will be an important topic for future work.

Moreover, we remain unconvinced that MEC just "is" a UGRNN-ReLU-Place Cell network, despite a network of this architecture having explained essentially all the data we had available to challenge it. It is possible that matching all our existing data is too easy a test. Would this network generalize to more complex situations, with increased variability along key axes such as spatial structure (e.g. environments with corridors and looping interconnections), nontrivial but spatially informative cues, and rewards exhibiting complex and temporally-variable patterns? We do not think it is at all obvious that it would. The proper conclusion from this work is thus not that our current best model is actually correct, but rather that the fairly simplified setup typical of experiments in MEC is not sufficiently complex to falsify it. This present situation is a concrete manifestation of the "contravariance principle" of neural modeling [Cao and Yamins, 2021b] – the idea that working in a *more* complex experimental environment might actually make it *easier* to identify an actually correct model, by virtue of reducing susceptibility to spurious models that are apparently consistent with too-simple data. Future experiments should engage mice in more complex environments and behaviors.

# 7 Acknowledgements

We are grateful to Alex Gonzalez, David Sussillo, and John H. Wen for helpful discussions. We thank the anonymous reviewers for their feedback on a draft of this manuscript. A.A. received support from the Swiss National Science Foundation (P2BSP3_181743 and P400PB_191076). I.L. is supported by funding from the Wu Tsai Neurosciences Institute under Stanford Interdisciplinary Graduate Fellowships and a Bertarelli fellowship. A.H.W. received funding support from the National Institutes of Health BRAIN initiative (1F32MH122998-01), and the Wu Tsai Stanford Neurosciences Institute Interdisciplinary Scholar Program. S.G. is supported by the James S. McDonnell Foundation, Simons Foundation, and National Science Foundation CAREER Award for funding while at Stanford. L.M.G. is supported by the Office of Naval Research N00141812690, Simons Foundation SCGB 542987SPI, the James S. McDonnell Foundation, and the Vallee Foundation. D.L.K.Y. is supported by the James S. McDonnell Foundation (Understanding Human Cognition Award Grant No. 220020469), the Simons Foundation (Collaboration on the Global Brain Grant No. 543061), the Sloan Foundation (Fellowship FG-2018-10963), the National Science Foundation (RI 1703161 and CAREER Award 1844724), the DARPA Machine Common Sense program, and hardware donation from the NVIDIA Corporation.

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
