# A  Comparisons to Neural Data

## A.1  Experimental Data

We analyze three neural datasets in total (two from tetrode recordings of freely moving animals in 2D arenas and one from Neuropixels recordings of head-fixed mice on a 1D virtual track). The 2D open-field foraging dataset in the $100cm^2$ arena came from Mallory et al. [2021] across 12 mice, totalling 620 MEC cells. The 2D open-field foraging (**reward-** condition) and reward dataset (**reward+** condition) in the $150cm^2$ arena came from Butler et al. [2019] across 7 rats, totalling 598 MEC cells. Note that the *same* neural population was used across these two conditions, whereby in the **reward+** condition, the animal navigates to a $20cm^2$ reward zone within this arena to receive 0.5-1 units of cereal. The reward zone location was fixed across sessions for each animal, but varied between animals. Finally, the 1D VR data was used by Mallory et al. [2021] across 8 mice, totalling 2861 MEC cells. There were five landmarks (towers) total at 0, 80, 160, 240, and $320cm$. For further experimental details, please refer to the respective cited paper referred above.

In the 1D VR data, we identified remapping events, depicted in Fig. S8, following the procedure used by Low et al. [2020], by examining the trial-by-trial similarity matrices as well as the test-set (90-10% splits) $R^2$ of $k$-means clustering (red) applied to the responses and identifying where it diverges from PCA (blue).

## A.2  Rate Map Representation

We bin the positions in each environment using 5 cm bins, following prior work [Hardcastle et al., 2017, Butler et al., 2019, Low et al., 2020]. Thus, the $100cm^2$ environment used 400 ($20 \times 20$) bins, the $150cm^2$ environment used 900 ($30 \times 30$) bins, and the $400cm$ 1D track used 80 bins. We then calculated each animal's binned occupancy in seconds as well as how many times each MEC cell spiked in each bin. Binned firing rates for each MEC cell were calculated as the ratio of the number of spikes and the time spent in that bin in seconds. Finally, a Gaussian filter ($\sigma = 1$ bin) with dimensionality matching the environment (1D or 2D) was used to smooth the rate maps.

Since the model units do not have spikes but rates, the procedure is analogous for the agent but without Gaussian smoothing. Specifically, to generate the model rate map (from layer $g$ for the path integrator networks, defined in (15)), we evaluate the model on 100 batches consisting of 200 evaluation trajectories (see Section C.1 for more details), with each episode being seeded to ensure the same evaluation trajectories are used across all models. Every timestep corresponded to $20ms$ increments of time.

See Fig. S5 for visualizations of the rate maps of both the neural data and the model.

## A.3  Mapping Transforms

When we perform neural fits, we choose a random 20% set of these position bins to train and cross-validate the regression, and the remaining 80% to use as a test set, across ten train-test splits total. For ElasticNet-based regression (including Lasso, Balanced Lasso-Ridge, and Ridge regression as special cases), we use the `sklearn.linear_model` class. When we perform cross-validation, which we do for the $100cm^2$ arena in 2D (Fig. 1b), the $400cm$ 1D track (Fig. 2b), and the $150cm^2$ for the 2D open foraging, reward, and combined across conditions datasets (Fig. 5b), we search over $\alpha \in [10^{-9}, 10^9]$ logspaced uniformly and L1 ratio spaced uniformly $\in [0, 1]$ with two-fold cross-validation on 20% of the position bins per train-test split and neuron individually. See Fig. S1 for the inter-animal consistency (2D data) across $\alpha$ and L1 ratio. For the 1D data, we choose the parameters on the validation set that perform best averaged across the maps for a neuron. The implementations of all of the transforms can be found here: `https://github.com/neuroailab/mec`.

## A.4  Grid Score and Border Score

We calculated each cell (and model unit) grid score by taking a circular sample of the spatial rate map autocorrelation centered on the central peak and compared it to rotated versions of the same circular sample ($60°$ and $120°$ versus $30°$, $90°$, and $150°$). The grid score [Langston et al., 2010, Butler et al., 2019] was defined as the mean correlation at $60°$ and $120°$ minus the mean correlation at $30°$, $90°$, and $150°$.

The border score was computed following Solstad et al. [2008], by calculating $\dfrac{CM - DM}{CM + DM}$, where $CM$ is the proportion of high firing rate bins along one wall, and $DM$ is the normalized mean product of the firing rate of each bin and its distance to the nearest wall.

The grid and border scores in Fig. 3 were always computed for each model at the layer of maximum neural predictivity (median across neurons). See Fig. S5 for visualizations of grid, border, and heterogeneous cells in both the neural data and the UGRNN-ReLU-Place Cell model.

# B    Neural Fitting Procedure and Inter-animal Consistency Definition

Suppose we have neural responses from two animals $A$ and $B$. Let $t_i^p$ be the "true" rate map of animal $p \in \mathcal{A} = \{A, B, ...\}$ on stimulus set $i \in \{\text{train}, \text{test}\}$ given by positions in the rate map. Of course, we only receive noisy observations of $t_i^p$, so let $s_{j,i}^p$ be the $j$-th set of $n$ trials of $t_i^p$. Finally, let $M(x; y)_i$ be the predictions of a mapping $M$ (e.g. Ridge regression) when trained on input $x$ to match output $y$ and tested on stimulus set $i$. For example, $M\left(t_{\text{train}}^{\text{A}}; t_{\text{train}}^{\text{B}}\right)_{\text{test}}$ is the prediction of the mapping $M$ on the test stimulus trained to match the true neural responses from animal $B$ given input from the true neural responses from animal $A$ on the train stimulus, and correspondingly, $M\left(s_{1,\text{train}}^{\text{A}}; s_{1,\text{train}}^{\text{B}}\right)_{\text{test}}$ is the prediction of the mapping $M$ on the test stimulus trained to match the (trial-average) of noisy sample 1 on the train stimulus from animal $B$ given inputs from the (trial-average) of noisy sample 1 on the train stimulus from animal $A$. Finally, $r$ is the rate map constructed from the model units, which by construction are deterministic across trials.

With these definitions in hand, we now define the inter-animal consistency and the model neural predictivity. Namely, when we have repeated trials in the same environment (as in the case of 1D data), we compute the following quantity for all units in target animal $B$:

Inter-animal Consistency$_{\text{1D}}^B :=$

$$\left\langle \frac{\text{Corr}\left(M\left(s_{1,\text{train}}^{\text{A}}; s_{1,\text{train}}^{\text{B}}\right)_{\text{test}}, s_{2,\text{test}}^{\text{B}}\right)}{\sqrt{\widetilde{\text{Corr}}\left(M\left(s_{1,\text{train}}^{\text{A}}; s_{1,\text{train}}^{\text{B}}\right)_{\text{test}}, M\left(s_{2,\text{train}}^{\text{A}}; s_{2,\text{train}}^{\text{B}}\right)_{\text{test}}\right) \times \widetilde{\text{Corr}}\left(s_{1,\text{test}}^{\text{B}}, s_{2,\text{test}}^{\text{B}}\right)}} \right\rangle_{A \in \mathcal{A} : (A,B) \in \mathcal{A} \times \mathcal{A}}, \quad (1)$$

Model Neural Predictivity$_{\text{1D}}^B :=$

$$\left\langle \frac{\text{Corr}\left(M\left(r_{\text{train}}; s_{1,\text{train}}^{\text{B}}\right)_{\text{test}}, s_{2,\text{test}}^{\text{B}}\right)}{\sqrt{\widetilde{\text{Corr}}\left(M\left(r_{\text{train}}; s_{1,\text{train}}^{\text{B}}\right)_{\text{test}}, M\left(r_{\text{train}}; s_{2,\text{train}}^{\text{B}}\right)_{\text{test}}\right) \times \widetilde{\text{Corr}}\left(s_{1,\text{test}}^{\text{B}}, s_{2,\text{test}}^{\text{B}}\right)}} \right\rangle_{A \in \mathcal{A} : (A,B) \in \mathcal{A} \times \mathcal{A}}, \quad (2)$$

where the outermost average is across all source animals $A$ that regress to the current target animal $B$, ten train-test splits, and 100 bootstrapped trials, Corr is Pearson correlation across test stimuli (in this case, held-out position bins), and $\widetilde{\text{Corr}}$ is Pearson correlation with Spearman-Brown correction applied to it, namely

$$\widetilde{\text{Corr}}(X, Y) := \frac{2\,\text{Corr}(X, Y)}{1 + \text{Corr}(X, Y)}.$$

In 1D, we can also have *multiple* maps within subsets of trials [Low et al., 2020], which we identify in Fig. S8. To account for this, we treat each map, $B^{m_i}$, which corresponds to responses of the same population in the target animal $B$ to a subset of trials, as its own target of explanation. The average over source animals $A \in \mathcal{A} : A \neq B$ in (1) now is an average over source animals and their respective maps, $A^{m_j}$. Note that in the standard limit of one map per animal, this is exactly the same as the original quantity in (1).

In the absence of repeated trials in the same environment (which was the case for the 2D data) and therefore also a single map per environment per animal, the terms in the denominator of (1) are trivially 1, giving us the following quantity for all units in target animal $B$:

$$\text{Inter-animal Consistency}_{\text{2D}}^B := \left\langle \text{Corr}\left(M\left(s_{\text{train}}^{\hat{A}}; s_{\text{train}}^B\right)_{\text{test}}, s_{\text{test}}^B\right)\right\rangle, \quad (3)$$

$$\text{Model Neural Predictivity}_{\text{2D}}^B := \left\langle \text{Corr}\left(M\left(r_{\text{train}}; s_{\text{train}}^B\right)_{\text{test}}, s_{\text{test}}^B\right)\right\rangle, \quad (4)$$

where now this outermost average is just across the ten train-test splits, and $\hat{A}$ is the sole "pooled" source animal constructed from the pseudo-population of the remaining animals distinct from the target animal $B$, namely, $A \in \mathcal{A} : A \neq B$. As expanded on in Section B.4, the latter use of the pooled source animal $\hat{A}$ is to ensure that we have a relatively comparable number of units in the source animal from the 2D tetrode data as in the 1D (Neuropixels) data, since otherwise there would be a small number of units in any single tetrode session.

The inter-animal consistency and model neural predictivity then is the concatenation (denoted by $\oplus$) of all of the inter-animal consistencies of all units in each animal $B \in \mathcal{A}$, over which we take median and s.e.m.:

$$\text{Inter-animal Consistency}_{\text{1D}} := \bigoplus_{B \in \mathcal{A}} \left\langle \text{Inter-animal Consistency}_{\text{1D}}^{B^{m_i}} \right\rangle_{\text{Maps } m_i \in B},$$

$$\text{Model Neural Predicitivity}_{\text{1D}} := \bigoplus_{B \in \mathcal{A}} \left\langle \text{Model Neural Predictivity}_{\text{1D}}^{B^{m_i}} \right\rangle_{\text{Maps } m_i \in B}, \quad (5)$$

$$\text{Inter-animal Consistency}_{2D} := \bigoplus_{B \in \mathcal{A}} \text{Inter-animal Consistency}_{2D}^{B},$$

$$\text{Model Neural Predictivity}_{2D} := \bigoplus_{B \in \mathcal{A}} \text{Model Neural Predictivity}_{2D}^{B}, \tag{6}$$

Thus, Figures 1b, 2a, and 5b are the median and s.e.m. of the quantities in (6). Fig. 2b is the quantity in (5), which we pass through $\tanh$ to be $\in [-1, 1]$, the same scale for visual comparison as (6), which we then compute median and s.e.m. over. Note that we compute these quantities in the 2D and 1D data at *each* model layer, and then report the performance at the layer of maximum median neural predictivity for each model.

In the following subsections, we give background on how the full quantity (1) can be obtained from the base "true" quantity we want to estimate (7). Note that these are not formal proofs (as they rely on assumptions which do not necessarily hold in all cases), but are meant to outline the motivations behind the final quantity.

## B.1 Single Animal Pair Motivation

The inter-animal consistency from one animal $A$ to another animal $B$ corresponds to the following "true" quantity to be estimated:

$$\text{Corr}\left(M\left(t_{\text{train}}^{A}; t_{\text{train}}^{B}\right)_{\text{test}}, t_{\text{test}}^{B}\right), \tag{7}$$

where Corr is the Pearson correlation across test stimuli. In what follows, we argue that this true quantity can be approximated with the following ratio of measurable quantities where we divide the noisy trial observations into two sets of equal samples:

$$\text{Corr}\left(M\left(t_{\text{train}}^{A}; t_{\text{train}}^{B}\right)_{\text{test}}, t_{\text{test}}^{B}\right)$$

$$\sim \frac{\text{Corr}\left(M\left(s_{1,\text{train}}^{A}; s_{1,\text{train}}^{B}\right)_{\text{test}}, s_{2,\text{test}}^{B}\right)}{\sqrt{\text{Corr}\left(M\left(s_{1,\text{train}}^{A}; s_{1,\text{train}}^{B}\right)_{\text{test}}, M\left(s_{2,\text{train}}^{A}; s_{2,\text{train}}^{B}\right)_{\text{test}}\right) \times \text{Corr}\left(s_{1,\text{test}}^{B}, s_{2,\text{test}}^{B}\right)}}. \tag{8}$$

In words, the inter-animal consistency corresponds to the predictivity of the mapping on the test set stimuli from animal $A$ to $B$ on two different (averaged) halves of noisy trials, corrected by the square root of the mapping reliability on animal $A$'s test stimuli responses on two different halves of noisy trials and the internal consistency of animal $B$.

We justify the approximation in (8) by gradually eliminating the true quantities by their measurable estimates, starting from the original quantity in (7). First, we make the approximation that

$$\text{Corr}\left(M\left(t_{\text{train}}^{A}; t_{\text{train}}^{B}\right)_{\text{test}}, s_{2,\text{test}}^{B}\right)$$

$$\sim \text{Corr}\left(M\left(t_{\text{train}}^{A}; t_{\text{train}}^{B}\right)_{\text{test}}, t_{\text{test}}^{B}\right) \times \text{Corr}\left(t_{\text{test}}^{B}, s_{2,\text{test}}^{B}\right). \tag{9}$$

by transitivity of positive correlations (which is a reasonable assumption when the number of stimuli is large). Next, by transitivity and normality assumptions in the structure of the noisy estimates and since the number of trials ($n$) between the two sets is the same, we have that

$$\text{Corr}\left(s_{1,\text{test}}^{B}, s_{2,\text{test}}^{B}\right) \sim \text{Corr}\left(s_{1,\text{test}}^{B}, t_{\text{test}}^{B}\right) \times \text{Corr}\left(t_{\text{test}}^{B}, s_{2,\text{test}}^{B}\right)$$

$$\sim \text{Corr}\left(t_{\text{test}}^{B}, s_{2,\text{test}}^{B}\right)^{2}. \tag{10}$$

Namely, the correlation between the average of two sets of noisy observations of $n$ trials each is approximately the square of the correlation between the true value and average of one set of $n$ noisy trials. Therefore, from (9) and (10) it follows that

$$\text{Corr}\left(M\left(t_{\text{train}}^{A}; t_{\text{train}}^{B}\right)_{\text{test}}, t_{\text{test}}^{B}\right) \sim \frac{\text{Corr}\left(M\left(t_{\text{train}}^{A}; t_{\text{train}}^{B}\right)_{\text{test}}, s_{2,\text{test}}^{B}\right)}{\sqrt{\text{Corr}\left(s_{1,\text{test}}^{B}, s_{2,\text{test}}^{B}\right)}}. \tag{11}$$

We have gotten rid of $t_{\text{test}}^{B}$, but we still need to get rid of the $M\left(t_{\text{train}}^{A}; t_{\text{train}}^{B}\right)_{\text{test}}$ term. We apply the same two steps by analogy though these approximations may not always be true (though are true for Gaussian noise):

$$\text{Corr}\left(M\left(s_{1,\text{train}}^{A}; s_{1,\text{train}}^{B}\right)_{\text{test}}, s_{2,\text{test}}^{B}\right) \sim \text{Corr}\left(s_{2,\text{test}}^{B}, M\left(t_{\text{train}}^{A}; t_{\text{train}}^{B}\right)_{\text{test}}\right)$$

$$\times \text{Corr}\left(M\left(t_{\text{train}}^{A}; t_{\text{train}}^{B}\right)_{\text{test}}, M\left(s_{1,\text{train}}^{A}; s_{1,\text{train}}^{B}\right)_{\text{test}}\right)$$

$$\text{Corr}\left(M\left(\text{s}_{1,\text{train}}^{\text{A}};\text{s}_{1,\text{train}}^{\text{B}}\right)_{\text{test}},M\left(\text{s}_{2,\text{train}}^{\text{A}};\text{s}_{2,\text{train}}^{\text{B}}\right)_{\text{test}}\right)$$
$$\sim\text{Corr}\left(M\left(\text{s}_{1,\text{train}}^{\text{A}};\text{s}_{1,\text{train}}^{\text{B}}\right)_{\text{test}},M\left(\text{t}_{\text{train}}^{\text{A}};\text{t}_{\text{train}}^{\text{B}}\right)_{\text{test}}\right)^2,$$

which taken together implies

$$\text{Corr}\left(M\left(\text{t}_{\text{train}}^{\text{A}};\text{t}_{\text{train}}^{\text{B}}\right)_{\text{test}},\text{s}_{2,\text{test}}^{\text{B}}\right)$$
$$\sim\frac{\text{Corr}\left(M\left(\text{s}_{1,\text{train}}^{\text{A}};\text{s}_{1,\text{train}}^{\text{B}}\right)_{\text{test}},\text{s}_{2,\text{test}}^{\text{B}}\right)}{\sqrt{\text{Corr}\left(M\left(\text{s}_{1,\text{train}}^{\text{A}};\text{s}_{1,\text{train}}^{\text{B}}\right)_{\text{test}},M\left(\text{s}_{2,\text{train}}^{\text{A}};\text{s}_{2,\text{train}}^{\text{B}}\right)_{\text{test}}\right)}}. \tag{12}$$

Equations (11) and (12) together imply the final estimated quantity given in (8).

## B.2 Multiple Animals

For multiple animals, we simply consider the average of the true quantity for each target in $B$ in (7) across source animals $A$ in the ordered pair $(A, B)$ of animals $A$ and $B$:

$$\left\langle\text{Corr}\left(M\left(\text{t}_{\text{train}}^{\text{A}};\text{t}_{\text{train}}^{\text{B}}\right)_{\text{test}},\text{t}_{\text{test}}^{\text{B}}\right)\right\rangle_{A\in\mathcal{A}:(A,B)\in\mathcal{A}\times\mathcal{A}}$$
$$\sim\left\langle\frac{\text{Corr}\left(M\left(\text{s}_{1,\text{train}}^{\text{A}};\text{s}_{1,\text{train}}^{\text{B}}\right)_{\text{test}},\text{s}_{2,\text{test}}^{\text{B}}\right)}{\sqrt{\text{Corr}\left(M\left(\text{s}_{1,\text{train}}^{\text{A}};\text{s}_{1,\text{train}}^{\text{B}}\right)_{\text{test}},M\left(\text{s}_{2,\text{train}}^{\text{A}};\text{s}_{2,\text{train}}^{\text{B}}\right)_{\text{test}}\right)\times\text{Corr}\left(\text{s}_{1,\text{test}}^{\text{B}},\text{s}_{2,\text{test}}^{\text{B}}\right)}}\right\rangle_{A\in\mathcal{A}:(A,B)\in\mathcal{A}\times\mathcal{A}}.$$

Typically, we may bootstrap across split-half trials and have multiple train/test splits, in which case the average on the right hand side of the equation includes averages across these as well.

Note that each neuron in our analysis will have this single average value associated with it when *it* was a target animal ($B$), averaged over source animals, bootstrapped split-half trials, and train/test splits. This yields a vector of these average values, which we can take median and s.e.m. over as we do with standard explained variance metrics.

## B.3 Spearman-Brown Correction

The Spearman-Brown correction is to be applied to each of the terms in the denominator individually, as they are each correlations of observations from half the trials of the *same* underlying process to itself (unlike the numerator).

## B.4 Pooled Source Animal

Often times, we may not have enough neurons per animal to ensure that the estimated inter-animal consistency in our data closely matches the "true" inter-animal consistency. In order to address this issue, we holdout one animal at a time and compare it to the pseudo-population aggregated across units from the remaining animals, as opposed to computing the consistencies in a pairwise fashion. Thus, $B$ is still the target heldout animal as in the pairwise case, but now the average over $A$ is over the sole "pooled" source animal $\hat{A}$ constructed from the pseudo-population of the remaining animals. We found that this pooling of the source animal units helped improve the estimated inter-animal consistency, as demonstrated in Fig. S7.

## C Model Training Details

All model code be found here: `https://github.com/neuroailab/mec`.

## C.1 Simulated Trajectories and Place Cell Representation

Place cell receptive field centers $\vec{c}_i$, $i = 1, \ldots, n_P$, distributed uniformly randomly across each environment. This environment is the $2.2m^2$ arena for all models except for the "Trained" bars in Fig. 2a, which corresponds to training in the $100cm^2$ environment, and the "1D Track" bar in Fig. 2b which corresponds to training on the $400cm$ track. We take $n_P = 512$ place cells in all environments and models, following Banino et al. [2018].

The response of the $i$-th place cell is simulated using a difference of Gaussians tuning curve, $p_i(x) = e^{-\|x-c_i\|_2^2/2\sigma_1^2} - e^{-\|x-c_i\|_2^2/2\sigma_2^2}$, where $x$ is the current location of the agent, $\sigma_1 = 0.12m$ and $\sigma_2 = 0.12\sqrt{2}m$. Agent trajectories are generated using the rat motion model of Erdem and Hasselmo [2012]. In 1D, during either training or evaluation we prevent the agent from making turns, in order to simulate the head-fixed condition that

the mice experience. We collect the place cell activations at $n_x$ locations as the animal explores its environment in a matrix $P \in \mathbb{R}^{n_x \times n_P}$.

## C.2 Place Cell Input Models

While the path integrator networks are trained with the place cells as supervised *outputs*, defined in Section C.3, we have several controls based on the place cell representation. The "Place Cell Ground Truth" model directly corresponds to the matrix $P$.

NMF corresponds to Non-negative Matrix Factorization on the matrix $P$, implemented via `sklearn.decomposition.NMF`. As noted by Dordek et al. [2016], this corresponds to a 1-layer neural network with $n_G$ hidden units via unsupervised Hebbian learning on inputs $P$, subject to a nonnegativity constraint. The "Grid Cell Model" corresponds specifically to NMF with $n_G = 9$ components, following Sorscher et al. [2019].

Finally, the "Place Cell Linear MEC" model is intended to be a neural data constrained linear alternative to NMF on the place cell matrix $P$. This is ElasticNet CV regression trained on 20% of the position bins in the current evaluation environment ($100cm^2$ arena in Fig. 2a and $400cm$ track in Fig. 2b), fitted to the neurons of animals distinct from the current target neural population (namely the units of the source animal defined in Section B).

## C.3 Loss Functions

The "Place Cell Loss" corresponds to the loss function used by Banino et al. [2018], which is the softmax cross-entropy loss between the ground truth timestep $t$ place cell targets $p_i^t$ and model outputs $\hat{p}_i^t$, given by

$$\mathcal{L}(\hat{p}, p) := -\frac{1}{T} \sum_{t=1}^{T} \sum_{i=1}^{N_p} p_i^t \log \hat{p}_i^t. \tag{13}$$

The "Position Loss" [Cueva and Wei, 2018] is given by

$$\mathcal{L}(\hat{p}, p) := \frac{1}{2} \frac{1}{T} \sum_{t=1}^{T} \left( \left( p_x^t - \hat{p}_x^t \right)^2 + \left( p_y^t - \hat{p}_y^t \right)^2 \right), \tag{14}$$

where $p_x^t$ and $p_y^t$ are the *Cartesian* coordinates $(x, y)$ of the agent's ground truth position at timestep $t$.

Both loss functions are averaged across the batch, where the path length in each batch for both loss functions is $T = 20$ timesteps. Additionally, for either loss function, we apply an L2 penalty of $1 \times 10^{-4}$ to the path integrator weights.

## C.4 Network Architectures and Hyperparameters

### C.4.1 Path Integrators

We use Tensorflow 2.0 for these models [Abadi et al., 2016]. The RNN path integrator network takes in 2 linear input units for $x$ and $y$ velocity, a set of recurrently connected input units, and linear readout units. The network update equations are as follows:

$$\begin{aligned} g^{t+1} &= f\left(Jg^t + Mv^t\right) \\ \hat{p}^{t+1} &= Wg^{t+1}, \end{aligned} \tag{15}$$

where $g$ is the vector of model MEC activities (4096 units total), $J$ is the matrix of recurrent weights, $M$ is the network's velocity input weights, $v$ is the agent's 2D velocity in the arena, $f$ is the element-wise nonlinearity (or the identity function if it is "Linear"), $\hat{p}$ is the vector of estimated place cell activities (or outputted Cartesian coordinates if the network is being trained with the "Position Loss"), and $W$ is the place cell (or Cartesian coordinate) readout weights.

The SRNN, GRU, LSTM, and UGRNN path integrator networks had an identical task and training protocol as (15). The model architecture was reproduced from Banino et al. [2018] except that our models had 4096 units (rather than 128) in order to match the number of units of the MEC layer ($g$) of the RNN path integrator above. Specifically, it consists of 2D velocity inputs to a recurrent circuit (SRNN, GRU, LSTM, or UGRNN) with 4096 units with nonlinearity either being Linear, Tanh, Sigmoid, or ReLU, followed by a nonlinear layer of 4096 units (which constitutes the model's activities $g$, with the nonlinearity matching that of the recurrent circuit, following Sorscher et al. [2019]), followed by a final readout to the estimated 512 place cell activities (or 2 Cartesian units of position if training with the "Position Loss"). Furthermore, the initial cell state and hidden state are initialized by computing a linear transformation of the ground truth place cells (or Cartesian positions if training with the "Position Loss") at time 0. We did not employ any dropout at the $g$ layer of these networks during training.

All networks are trained with Adam [Kingma and Ba, 2015] with a learning rate of $1 \times 10^{-4}$, batch size of 200, and 100 training epochs consisting of 1000 batches of trajectories per epoch.

### C.4.2 Cue Input

MEC neurons are known to respond to visual input, but as none of our evaluated models had a visual front-end, they could not respond accordingly when evaluated on a cue-rich environment (which was the case for the 1D data). To test this hypothesis, we trained the UGRNN-ReLU-Place Cell model in the $2.2m^2$ 2D arena, but with input visual cue locations concatenated with the 2D velocity input, corresponding to the "Cue Input + Velocity Input" model in Fig. 2b. Specifically, these visual cue locations were a vector $\ell \in \mathbb{R}^5$, corresponding to 5 cues placed in fixed, arbitrary locations in the 2D arena with widths between $0.06$ to $0.3m$ on each side. Each element $\ell_i$ of this cue input vector corresponds to the Euclidean distance of the agent at current time $t$ to the nearest boundary of the $i$-th cue (and 0 for that entry if the agent is within the boundaries of this $i$-th cue). We also considered a UGRNN-ReLU-Place Cell network trained in 2D without any velocity input and only the cue input, corresponding to the "Cue Input Only" bar in Fig. 2b.

Finally, we evaluated these networks with the 1D cue input which matched the widths and locations of the cues in the 1D virtual track. The neural predictivity of this 1D cue input is the "Cue Input" bar in Fig. 2b. As a control, we also included a UGRNN-ReLU-Place Cell path integrator trained with the usual 2D velocity input but with constant 0 concatenated to the velocities in place of the active cue input (thus being functionally equivalent to the path integrators in Section C.4.1). The network was then provided the 1D cue inputs during evaluation only, corresponding to the "Evaluation Cue Input Only" bar in Fig. 2b.

### C.4.3 Reward Biased Path Integrator

For the reward biased path integrator (parametrized by $\varepsilon$ in Fig. 5b), we trained the UGRNN-ReLU-Place Cell in the $2.2m^2$ arena, where on $(1 - \varepsilon)$ fraction of training batches (each example in the batch corresponding to a 20 timestep path-length episode), we had the agent navigate directly within a fixed number of timesteps (7) to the center of a $20cm^2$ reward zone placed in an arbitrary, fixed location of the environment. In the remaining 13 timesteps of these episodes, the agent path integrated using the motion trajectories of [Erdem and Hasselmo, 2012] but restricted to the $20cm^2$ reward zone. In the other $\varepsilon$ fraction of episodes, the trajectories were unchanged from before.

As a control, we consider a "Reward Input" path integrator (Fig. 5b,c), which does *not* employ the reward biased trajectories (so $\varepsilon = 1$), but instead takes an additional scalar reward signal (concatenated with the 2D velocity input) during training, indicating if it is in the reward zone or not at current timestep $t$.

## D  Performance Measure and Ablation

Positional decoding error for the place cell loss models is measured first by computing a predicted Cartesian position $\hat{p}_x^t, \hat{p}_y^t$, obtained by taking the top 3 most active place cell outputs at each timestep in each model and averaging them. Finally, for each trajectory episode (of length $T = 20$), we compute the following measure of path integration error,

$$\mathcal{E}(\hat{p}, p) := -\frac{1}{T} \sum_{t=1}^{T} \sqrt{\left( (p_x^t - \widehat{p}_x^t)^2 + (p_y^t - \widehat{p}_y^t)^2 \right)}, \tag{16}$$

where the error is additionally averaged over the batch dimension (20,000 examples computed from 100 batches of 200 evaluation trajectory episodes each).

The performance degradation metric that is the $y$-axis of Fig. 3d is given by $\left( \mathcal{E}^{curr} - \mathcal{E}^{full} \right) / \mathcal{E}^{full}$, where $\mathcal{E}^{full}$ is the performance of the trained UGRNN-ReLU-Place Cell model, and $\mathcal{E}^{curr}$ is the performance of the same model but with a subset of its population of units in the $g$ layer set to 0. In Fig. 4d, for the "grid cell knockout", the outputs of the set of units in layer $g$ with grid score $> 0.3$ are set to 0 during evaluation, and for the "heterogeneous cell knockout", the same number of units in layer $g$ with grid score $\leq 0.3$ are randomly set to 0 (subsampled 100 times). Analogously, for the "border cell knockout", the border score was set to a threshold of 0.5, and the heterogeneous knockout in that case was border score $\leq 0.5$.

# E    Supplementary Figures

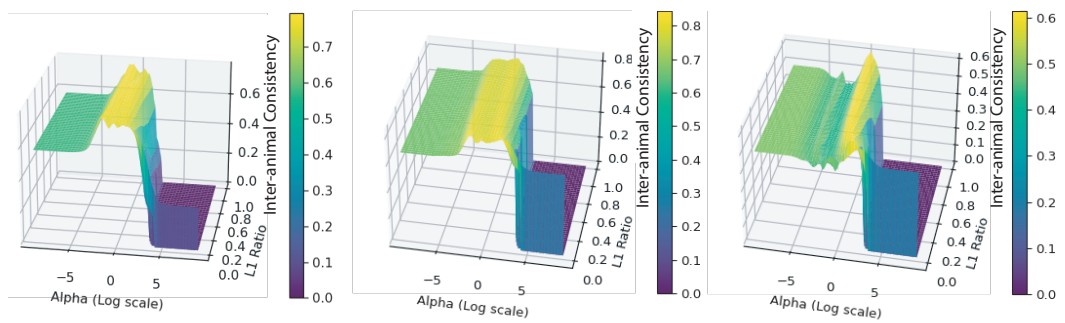

Figure S1: **Inter-animal consistency as a function of $\alpha$ and L1 ratio (sparsity penalty strength).** For $\alpha \in [10^{-9}, 10^9]$ logspaced uniformly and L1 ratio spaced uniformly $\in [0, 1]$, we plot the inter-animal consistency evaluated on 80% of position bins for a given train-test split and cell on the $100cm^2$ arena.

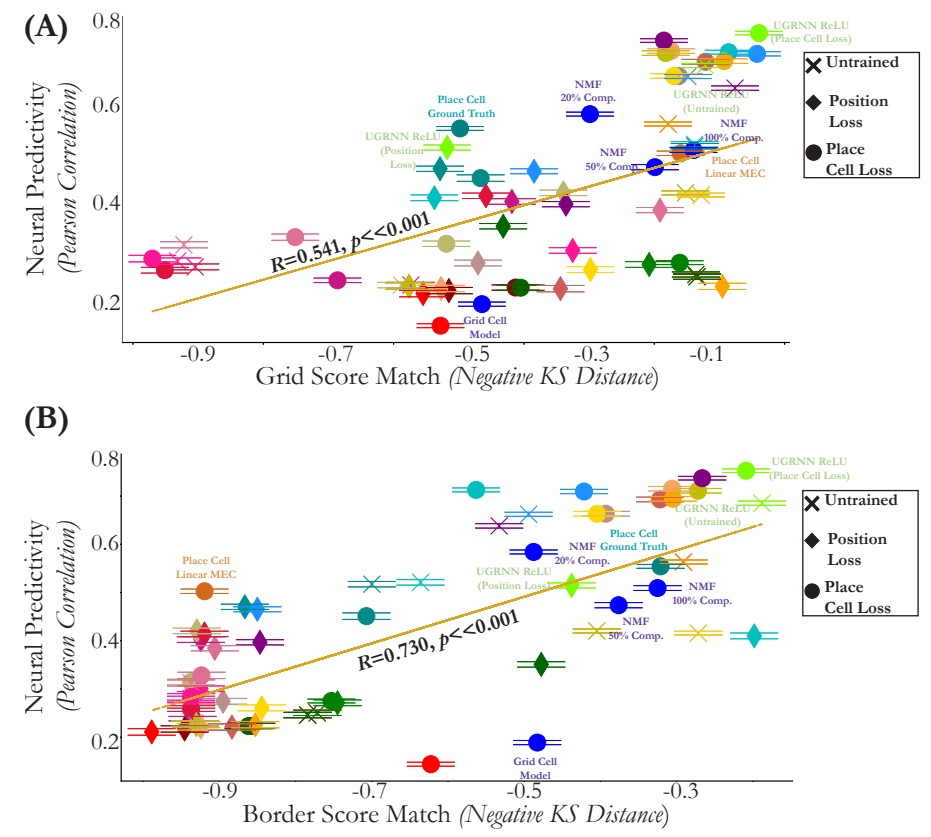

Figure S2: **Neural predictivity and grid & border score distribution match are related.** Each model's neural predictivity on the $100cm^2$ foraging data versus grid score (top) and border score (bottom) distribution match. The neural predictivity is the median and s.e.m. across 620 cells. The models are either untrained ("X"), trained with the place cell loss ("O") in the $2.2m^2$ arena, or trained with the position loss ("Diamond") in the $2.2m^2$ arena.

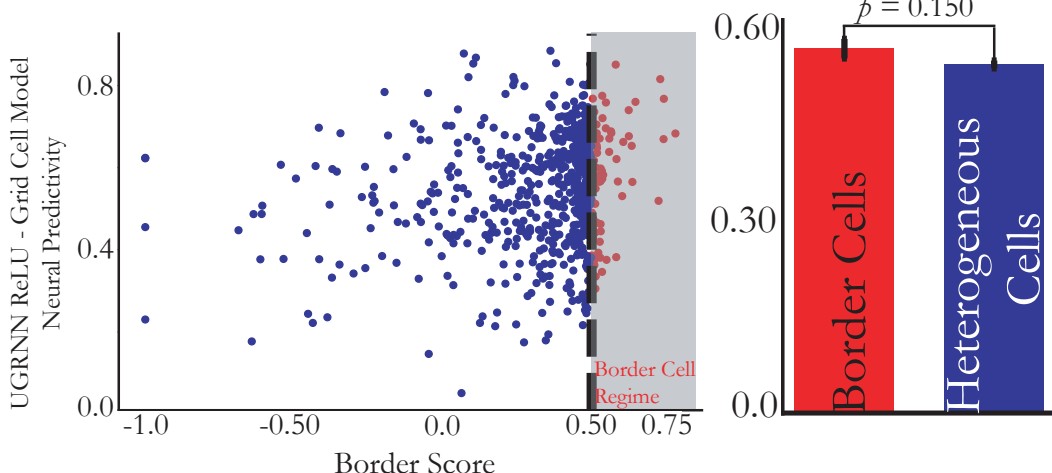

Figure S3: **Difference between UGRNN-ReLU-Place Cell and Grid Cell models (Border Score).** (*Left*) Per unit neural predictivity difference between the UGRNN-ReLU-Place Cell and Grid Cell models trained on the $2.2m^2$ arena and evaluated on the $100cm^2$ arena, plotted against that unit's border score. (*Right*) Quantification of this difference aggregated across the border cell and heterogeneous (non-border) cell populations, respectively (mean and s.e.m).

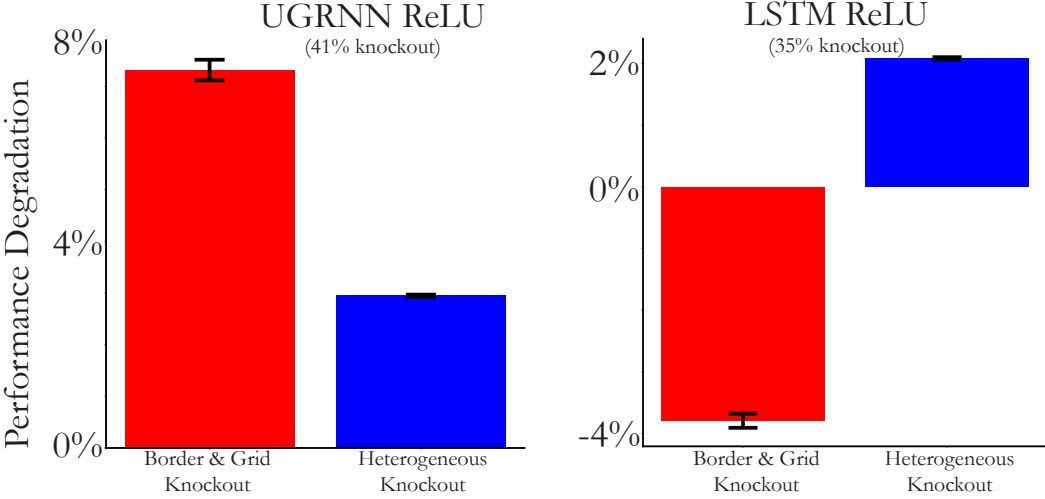

Figure S4: **Combined border and grid cell knockout.** For the UGRNN-ReLU-Place Cell and LSTM-ReLU-Place Cell models trained on the $2.2m^2$ arena, we select grid cells (grid score > 0.3) and border cells (border score > 0.5) to knockout (red), and knockout the same number of heterogeneous cells (neither border nor grid cell), randomly sampled 100 times. The percentage knockout refers to the percentage of total units knocked out in the current layer, corresponding to the intermediate layer of the three layer network. Mean and s.e.m. over evaluation episodes.

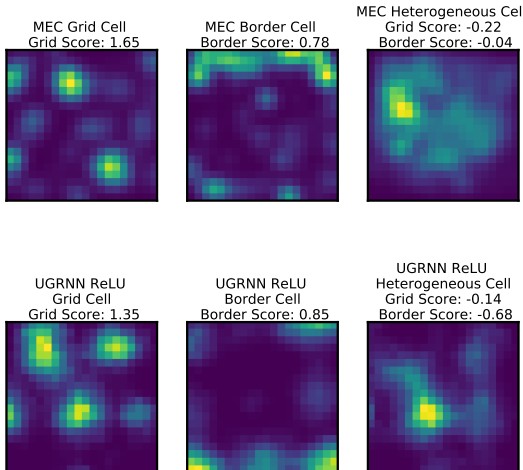

Figure S5: **Example rate maps.** Example rate maps obtained from animals foraging in the $100cm^2$ arena (top row) and rate maps from the UGRNN-ReLU-Place Cell trained in the $2.2m^2$ arena and evaluated on the $100cm^2$ arena. We also include the grid and border scores of these example units.

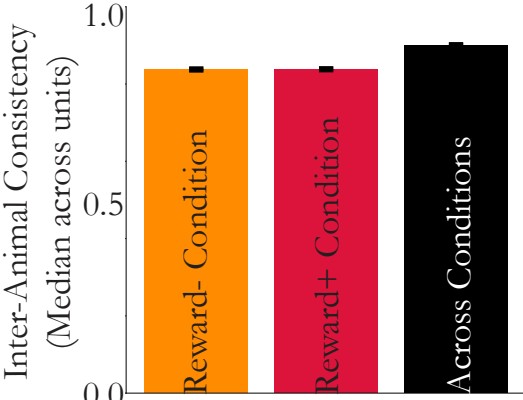

Figure S6: **Inter-animal consistency is high per condition and across conditions.** Inter-animal consistency of neural responses (under ElasticNet CV regression) is computed per **reward-** and **reward+** condition separately, and across both spatial and reward-modulated conditions overall. Median and s.e.m. across 598 MEC cells from 7 rats in the $150cm^2$ arena.

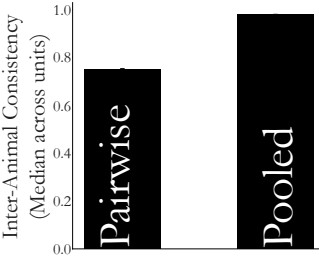

Figure S7: **Pooling across source animals.** Inter-animal consistency under Ridge regression ($\alpha = 1$) trained with 50% position bins (averaged across ten train-test splits) from 12 mice foraging in the $100cm^2$ arena. Median and s.e.m. across 620 cells. "Pooled" refers to computing the inter-animal consistency using a single "pooled" source animal from units gathered from all animals except the target animal, as described in Section B.4. "Pairwise" refers to computing this quantity mapping one source animal at a time to the target animal.

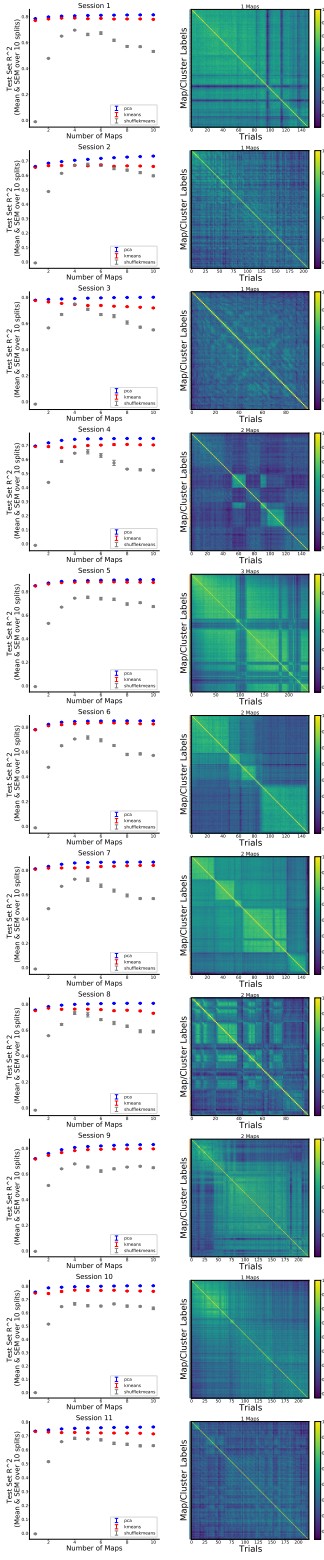

Figure S8: **Remapping analysis for 1D VR data.** For each recording session (11 total), we examine the trial-by-trial similarity matrices (right column) and the test set $R^2$ of $k$-means clustering as a function of $k$ (left column) relative to both PCA (blue) and a shuffled control (gray). The number of identified maps per session is listed at thee top of each trial-by-trial similarity matrix, along with the colored cluster labels as to the assignments for each trial.