# OpenReview forum: "Explaining heterogeneity in medial entorhinal cortex with task-driven neural networks"
_NeurIPS.cc/2021/Conference — NeurIPS 2021 Spotlight_

### Official Review · Reviewer_GE9u · 2021-07-07

**Rating:** 7
**Confidence:** 4

**Summary:**

The authors used linear regression to first demonstrate the robustness of non-grid-like responsive cells in experimental recordings of MEC cells. They then used the same regression to assess the similarity of responses derived from the goal driven training of RNNs to MEC responses. They found that responses derived from RNNS contained these non-grid-like responses and explained neural activity better than a low-rank decomposition of place fields.  These non-grid-like responses were crucial for path integration and likely explained the difference in similarity matching between goal-driven training and low rank decomposition.

**Limitations And Societal Impact:**

Not applicable.

**Main Review:**

There are significant pros and cons with this paper currently. It is written clearly and each part is well-motivated. Adequate controls were performed, and is fairly rigorous.  However, the central finding is simply that goal-based training provides a better fit to real data as compared to a specialized grid cell model. There are some hints that this gap in performance arises from responses that are not easily categorized (Figure 3+4), but there is little insight over the nature of these response types -- are they border cells, heading cells, mixed selectivity cells, etc. I would like to know what these response types are and why they improve similarity matching. I feel that a further exploration of these response types, as detailed in the comments, will greatly strengthen this paper, so I will await author responses.

Comments:

- The main contribution of the paper rests upon understanding cells that cannot be categorized classically into the typical navigation variables such as border / grid / head direction cells. However, there is a clear distinction between border / head direction cells and cells with heterogeneous of mixed selectivity (i.e. cells with both heading and grid responses). But are all of these types of responses treated as heterogeneous cells because they have low grid scores? If so, the definition supplied in the introduction does not agree with the definition of heterogeneous cells in the results section. Shouldn't border, heading, and heterogeneous cells be treated independently?

- It is suggested that "heterogeneous" cells, operationally defined as having low grid scores in figure 1, are responsible for the performance difference between RNN-based goal-driven training vs NMF (Figure 4C). If you take these cells out, is there less of a difference between NMF vs goal-driven approaches? How does the decrease in the performance gap compare to if border cells or head direction cells were taken out? Put another way, is the failure of NMF in reproducing neural responses simply because it can't reproduce border and heading responses, or because it can't reproduce mixed selectivity responses?

- It's unclear how many of these heterogeneous cells exist in the data. Even if this is reported previously it should still be shown here to provide context. Figure 1c also does very little justice in showcasing that heterogeneous cells constitute a unique functional population.

- I have some concerns over the notion of similarity. For one-to-one mapping, can neuron A1 in animal A be mapped, with replacement, to multiple neurons, B1 and B2, in animal B if A1 is the best match to B1 and B2? If this is allowed, then hypothetically, animal A, despite having only a single grid cell and 99 head direction cells, may actually serve as a good template / have a high similarity using ridge regression to animal B, which has 100 grid cells as long as a sufficient basis set of grid, head, border, etc cells exist in every recorded animal. Perhaps the current notion of similarity does not factor in differences in the relative proportion of unique responses across recorded animals? If so, the notion of similarity in the paper is different from a neuroscientists' notion of similarity when viewing population PSTHs of each animal, and may need some degree of qualification in the text.

- Some simple negative controls are needed to interpret similarity measures. For instance, similarity should not be high when regressing a classical grid response when exclusively using classical head direction cells, and vice versa and also for using other classical response types, such as border cells, etc. The similarity levels will also provide the reader with some baseline to evaluate what the numbers mean in figure 1B.

- I may be reading figure 1C wrong, but it appears that target unit predictivity is marginally higher for het cells than grid cells when using a smaller fraction of source population. Doesn't this mean that het cells actually require fewer source cells to achieve high similarity scores?

- The legends to the different sub-bars in Figure 2 should be made clearer.

- It's unclear to me what the inclusion of reward responses and Figure 5 adds to the central findings of paper.




**Time Spent Reviewing:**

4

---

> ### Author Response · Authors · 2021-08-10
> **Response to Reviewer GE9u (Part 1 of 2)**
>
> - There are some hints that this gap in performance arises from responses that are not easily categorized … [but] are they border cells, heading cells, mixed selectivity cells … I would like to know what these response types are.... The main contribution of the paper rests upon understanding cells that cannot be categorized classically into the typical navigation variables … But are all of these types of responses treated as heterogeneous cells because they have low grid scores? … Shouldn't border, heading, and heterogeneous cells be treated independently? …. It's unclear how many of these heterogeneous cells exist in the data. …
>
> *To clarify, by “heterogeneous”, we just mean cells with low grid scores.  We find that approximately 23% of our cells (145/620) are grid cells (grid score > 0.3), so the remaining 77% are “heterogeneous” in this sense.  Following the reviewer’s suggestion to look at border cells as well, using a standard metric of border-ness (Solstad et al. 2008), we find that approximately 12% of cells (78/620) are border cells, so that a total of 65% of units are neither grid nor border cells. We now call these non-grid non-border cells “extra-heterogeneous” (happy to use a better terminology if suggested!)*
>
> *We’ve also quantified the relationship of these categories to head-direction selectivity.  Using the definitions from Sargolini et al. 2006, we find that a large fraction (61.1% = 379/620) of our cells are head-direction selective.  The majority of these are mixed-selectivity, as very few of these are pure head-direction cells.  We find, however, no interaction between being head-direction selective and being heterogeneous -- the fraction of heterogeneous cells that are head-direction selective is essentially the same as in the population as a whole (62% = 295/475).  Thus, it is probably best to think of the het cells as an independent cross-cutting distinction to mixed-selectivity.*
>
> *Just to point the reviewer toward the relevant comparison to models, in the original submission we evaluated the match between models and real data on the full distribution of grid scores.  The best models match these almost exactly, which apply that they have the right percentage of grid cells (since the grid-score distribution is strictly more information).   In response to the reviewer’s question, we’ve also (as described below in more detail) quantified the same thing for border cells.*
>
> - It is suggested that "heterogeneous" cells, … are responsible for the performance difference between RNN-based goal-driven training vs NMF (Figure 4C). If you take these cells out, is there less of a difference between NMF vs goal-driven approaches?
>
> *Hm, slightly confused.  In our understanding, this is what 4C already shows, namely that  the gap between the RNN model and the NMF model is substantially lower for high-grid-score cells. This is consistent with previous works showing the NMF model is not a bad model for high-grid-score cells [Dordek et al. 2016; Sorscher et al. 2019].  We suspect we probably have somehow misunderstood this question but just aren’t sure exactly what to do. Sorry!*
>
> - How does the decrease in the performance gap [Fig 4D] compare to if border cells or head direction cells were taken out?
>
> *This is a good question!  In response to the reviewer’s question, we have knocked out the “extra heterogeneous” cells (non-grid-non-border), and compared this to knocking out the combination of grid+border cells.  We find that the gap between the “extra-heterogeneous cell” knockout and the “grid+border cell” knockout still exists, and is approximately the same magnitude as with the earlier grid-only knockout.*
>
> - Figure 1c also does very little justice in showcasing that heterogeneous cells constitute a unique functional population.
>
> *Our analyses indicate that there’s a continuum between “strictly grid-like” and “highly heterogeneous” -- rather than a strong segregation of the population into two subsets.  If the latter were true, we might expect the empirical grid-score distribution (gray bars in 3A) to be bi-modal, but it’s definitely not.  Similarly, we don’t find that we need two different model types to explain the low- and high- grid-score cells. Instead, the same RNN model that is best for one end of the grid-score spectrum is actually also the best for the other.
> To the extent we talk about the het/grid distinction at all, we do so more a way to emphasize that a whole bunch of cells whose spatial responses had been largely ignored in the literature can now be modelled effectively -- and that these cells are likely to be just as (if not slightly more) relevant to functionality as grid cells are.   But we do not mean to imply these cells are a distinct functional sub-population.  We will make this clearer in the discussion section in revision.*
>
> - I have some concerns over the notion of similarity. ... Perhaps the current notion of similarity does not factor in differences in the relative proportion of unique responses across recorded animals? ...
>
> *Yes, exactly!  This is indeed a “problem” with the regression notion of similarity. As long as two populations (the “source” and the “target”) contain the exact same types of units, the prevalence in the population of each type will not matter.  The observation that many models still do poorly on the regression metric is evidence of the fact that those models don’t even have the right types units at all -- having the right types of units (independent of their exact prevalences) turns out to be a strong constraint on model correctness.*
>
> *But the fact that the regression metric is prevalence-insensitive is exactly the reason why we also include an additional metric in addition to that of regression similarity, namely match of the grid-score distribution (see Fig. 3).  That metric is directly sensitive to the actual prevalence of unit types in the population.  For a model to do well at this metric, it must have the right proportions of units in each bin of the distribution.   (Of course, this metric is more limited than the regression metric in terms of the ways it checks similarity between the models and the data, simply because it only considers things in terms of the grid-ness).   The best of our models do well on both metrics.  That means such a model not only has the right types of units in a general sense but also the right proportions of units of each (grid-bin) type.*
>
> *Actually, in response to the reviewer’s question about border cells, we have now also evaluated a border-score distribution match metric, in analogy to what we did previously with grid cells.  We find similarly that the best models have excellent (though not quite perfect) border cell distribution match even without any regression, and the rank order of correctness across models here is essentially the same as for the grid-score matches.  We will include these results in supplementary figures to the revision.*
>
> *One other thing to keep in mind, however, is that the “problem” of prevalence-insensitivity with the regression metric might in some situations be more a “feature” than a “bug”.   It is quite plausible that two animals will themselves differ in the prevalence of various unit types, even when the set of those types is similar.  In that case, we actually *want* the metric establishing similarity between animals to not be too sensitive to prevalence levels, since a metric that does enforce pure prevalence sensitivity would lead to an artificially low inter-animal consistency (and thus artificially low noise ceilings for models to match).  Such a situation comes up in macaque V1 cortex, where two animals often have different fractions of cells with a given orientation selectivity. A strictly prevalence-sensitive metric would (spuriously) suggest that these animals don’t have functionally similar V1 cortices.   It is an open question as to whether this situation actually does come up here in the MEC context, but it is something we seek to explore in future work.*
>
> - Some simple negative controls are needed to interpret similarity measures ....
>
> *Good question.  We performed additional analyses predicting heterogeneous and grid cells with border cells and vice versa.  We find that within each classical cell type, predictivity is higher than for that cell type to the other cell types or to the population as a whole.  All of these numbers are substantially below the inter-animal consistency numbers.  Taken together, these results suggest that the regression metric doesn’t trivially conflate differences between the classical cell populations.  We will add these results in supplementary figures to the revised paper.  As for comparisons to/from head-direction-cells, since a large fraction of our cells exhibit head-direction selectivity, but only a very small number are purely head-direction selective rather than just being mixed-selective, there wasn’t really a segregated classical head-direction population to use as a control, so we did not do this particular comparison.*

---

> > ### Author Response · Authors · 2021-08-10
> > **Response to Reviewer GE9u (Part 2 of 2)**
> >
> > - I may be reading figure 1C wrong, but it appears that target unit predictivity is marginally higher for het cells...
> >
> > *Yes, exactly!  But why do we see this?*
> >
> > *One possibility is that our regression method could have been at fault, e.g. for each grid cell target, there is a matching grid cell in the source, but the regression is just not finding it.  We ruled out this possibility in two ways:  first, the 1-to-1 mapping method we tried would have picked it up, were it to exist; but it did not.  Also, we performed an analysis using mock data with pure populations of grid cells; we find that if you actually had two large random populations of pure grid cells, the regression method ends up putting all its weight on essentially the one right source cell for any target.  (We will add this analysis as a supplementary figure in the revised version of the paper.)*
> >
> > *Instead, the likely reason for this observation is that the actual distribution of grid cells between animals is not more self-similar than that of the het cell population.   We intuitively have an idealized picture in our heads in which MEC contains grid cells of a wide range of nicely-distributed (if quantized) spacings, orientations, and phases.  However, the idealized picture might be wrong.  It might simply not be the case that the grid-cell populations in one animal are all that well matched to grid-cell populations in other animals, at least no more so than the non-grid cells.*
> >
> > - It's unclear to me what the inclusion of reward responses and Figure 5 adds to the central findings of the paper.
> >
> > *One of the key types of findings of the past few years is that MEC is not “just” a pure navigation area.   Instead, it’s responses are modulated by things that are non-spatial and abstract (including reward).  Thus, it is a very natural question as to whether these non-spatial response properties are easily encompassed within the modeling framework that we had established.  Without the reward modeling we showed, one could easily object: “ok your idea works for spatial path integration, but doesn’t have anything to say about all these new results about MEC”.  After all, this very same criticism **would** be quite valid if leveled at classic grid cell models (or the NMF model for that matter) -- which it and of itself exemplifies the limitation of such hand-tuned purpose-specific models and one of the main potential benefits of task-driven models.*
> >
> > *The fact that essentially the same modeling framework also captures reward responses shows how the conclusions of the previous sections about task and architecture robustly generalize to what otherwise might be thought of as a very different domain.   Actually, a concrete example of how this is useful arose in response to the comments of reviewer VFHU above (see the question beginning “Is there any possibility that differences in the rate maps between…” and our answer to that question).   Here the reviewer is asking about whether the models can help shed light on the question of whether reward-bending of grid cell responses is partially dependent on explicit reward-input drive to the MEC system, or just a response to changes in velocity profiles of the animals.  Because our reward-prediction models are essentially the input-output structure as the previous model, but now forced to deal with differing velocity profiles -- and this difference seems enough to explain reward-bending of grid-cell neural responses nearly perfectly -- this does seem to shed light on the reviewer’s question (especially since the alternate explicit-reward input model doesn’t work nearly as well).  In response to your question, the key point is that, by tying a number of apparently disparate domains together into a simple overall modeling framework, many questions (even ones we didn’t at first think of) can be addressed that might have otherwise been harder to tackle.*

---

> > > ### Comment · Reviewer_GE9u · 2021-08-17
> > > **response**
> > >
> > > I thank the authors in answering my previous observations with such attention and care. My main concern, which is the characterization of other distinct cell classes in the MEC (border, heading, etc) are assuaged by what appears to be a rather thoroughly conducted characterization of these prevalent classes. The knockout of the 'extra-heterogeneous' cells (I leave it to the authors to come up with a simpler name)  along with the negative controls are also reassuring. I have modified my rating accordingly to reflect the improved rigor and interpretability of these results.
> > >
> > > There is still an important question that I believe is left answered, but please correct me if there are already plots addressing this issue. I realize this may be a very hard question to answer, but any insight, even absent experiments, should be useful. We know now that these extra heterogeneous cells are neither grid or border cells. We also know, according to Figure 4, that they contribute to path integration. But currently the definition of this het population is a definition of exclusion. Do we have insight on what these cells are actually encoding? Are the heterogeneous cells as heterogeneous relative to each other as compared to the heterogeneous cells found in the MEC? Can they be regressed against other variables available in the data or do they truly evade a satisfying classification after performing a GLM analysis?

---

> > > > ### Author Response · Authors · 2021-08-19
> > > > **response to Reviewer GE9u's response**
> > > >
> > > > *We thank the reviewer for their careful consideration of our responses and for updating their score!*
> > > >
> > > > - We know now that these extra heterogeneous cells are neither grid or border cells. ... But currently the definition of this het population is a definition of exclusion. Do we have insight on what these cells are actually encoding? Are the heterogeneous cells as heterogeneous relative to each other as compared to the heterogeneous cells found in the MEC? Can they be regressed against other variables available in the data or do they truly evade a satisfying classification after performing a GLM analysis?
> > > >
> > > > *This is a good question.  We agree that the fact that the heterogeneous population is defined just by exclusion, as you say, is somehow not an optimal situation.  Ideally we could do just as you recommended, which is to explain most of these cells' responses by regressing from state variables available in the data.  We've tried to do this from all the state variables we have in the data and haven't found much.  To be honest, there aren't that many position-independent state variables we have in the first place -- just running speed and head-direction really.  The main regression we've implemented here is the "input control" in the left-most bar of Fig 2a.  This is a regression of responses form the combined speed-head direction vector, and this control doesn't itself explain responses well.*
> > > >
> > > > *Where does this leave us? In a sense, our best current characterization of the het cells is that they are the things that are needed for a circuit architecture of the "right kind" (e.g. gated-recurrent-with-rectification, as in the LSTM-ReLU model) to achieve the "right task" (e.g. place field construction).  This is a characterization of the responses not in terms of simple state variables, but in terms of which structural circuit type the system uses and which functional constraint "goal" the system is constrained by.*
> > > >
> > > > *The alternative hypotheses ruled out by such an analysis are other possible circuit structures (e.g. without rectification or gating) and other possible task types (e.g. direct position encoding).  These are ruled out by virtue of being bad explainers of the data.  Thus, this ends up being a kind of indirect inference about which circuits / tasks are correct.
> > > > Of course, this doesn't really answer your original question, but proposes that in lieu of getting direct answers to your question, maybe we should kind of shift the question a little. Ultimately, we agree that a more satisfying situation would be one where we had some additional state variables that were easily interpretable and which explained the responses well, but at the moment we haven't been able to figure something like that out yet. It may turn out, moreover, that doing so is very hard or impossible, and that the most effective characterizations end up being in terms of the constraints on the network that can describe the data rather than in terms of easily stated single-unit-level classifications. Ideally, one would have characterizations of both kinds, but it's not clear how far that will be possible.*

---

### Official Review · Reviewer_PoER · 2021-07-13

**Rating:** 6
**Confidence:** 4

**Summary:**

The authors address two important questions regarding the non-grid cells in MEC. First, is their response consistent, and not just noise? Second, can they be used as criteria for testing models of MEC functionality? The second question allows to probe the functional significance of these cells, and the effect of incorporating them on various model properties.
The authors’ main tool is a linear mapping from a population of cells into a single target cell. This is done using rate maps of cells, and cross validation on the spatial bins. Several existing models are compared, with one model singled out as corresponding best to experimental data.
The functional relevance of grid and non-grid cells is evaluated using ablation studies on the model, where path integration ability is used as a measure. Generalization of the models is assessed using either different arenas or a reward-biased setting.



**Limitations And Societal Impact:**

Societal impact – not relevant.
Limitations: mentioned above.


**Main Review:**

**EDIT** *After carefully reading all the reviews and author responses, I am raising my score to a 6.*


I think these are important and timely questions. The idea of comparing firing patterns across animals and between animals and models is a good one. It has been used in other studies, but I think not many compared between-animal fit as a reference point. Checking the interplay between fitting the entire population and functional aspects is also a promising approach.
Unfortunately, as I detail below, the methods and analyses used by the authors do not adequately answer the questions or support the conclusions of the paper.
The main tool used is consistency between animals and models. I found very little explanation of the motivation for this choice of metric, and it was also quite hard to understand the details of the process. The authors chose a linear mapping from a population of neurons into a single neuron. Crucially, the train and test data are rate maps and the test data are 50% of the bins, under a random choice. The rate maps are also smoothed with a 1-bin wide Gaussian. This means that the test data is not that different from the train data. Given this procedure, I was actually surprised that not all mappings resulted in very high scores (although the animal-to-animal scores were). I can only assume that something about the statistics of the maps under different architectures is responsible, but this is not detailed or explored. Furthermore, the authors ask on line 120 “What firing patterns of MEC cell populations are common across multiple animals?” This is a great question, but I don’t see how the analysis in the paper answers it. It could be that one animal has grid cells and another animal has stripe cells, but they can be linearly combined to give one another.
Another major tool used is the gridness score. A threshold of 0.3 (typical for the literature) is used to divide the cells, and then compare the different populations. The distributions of gridness score, however, show a wide unimodal distribution with nothing special happening at 0.3. It would strengthen the conclusions if the authors check a range of thresholds, or even omit the cells with a gridness score close to 0.3 (for instance only analyze the bottom and top quadrants).
As for clarity – reading the figures was very challenging. The legends and captions do not provide a complete explanation, and the figures are extremely dense.
There is related literature on the importance of heterogeneous cells that should be probably discussed: Meshulam et al, Neuron 2017; Stefanini et al, Neuron 2020
Specific points:
1.	Line 13. “that that”
2.	Lines 52-62. The choice for the similarity transform should be better explained.
3.	L100 “solves the neurons”. Not clear when reading this
4.	Supplementary: the transform is not explained clearly enough, given its centrality in the paper.
5.	Supplementary: All models have the same trajectory. Animals do not.
6.	Supplementary: 50% of the spatial bins is quite a dense sample. In general, using different trajectories, and actual responses of the neurons (instead of rate maps) might be a more fair comparison.
7.	L136: “between the population rate maps”. This should be stressed (using rate maps and not responses to specific trajectories), and introduced earlier in the paper
8.	Figure 1A : mirror text
9.	Figure 1C: the boundary at 0.5 is confusing. Not clear what it refers to.
10.	Figure 2: Extremely hard to read. The caption doesn’t provide all the information.
11.	L180: models are substantially different. How consistent is the pattern of figure 2 across different animals? Error bars are across 2861 units. It would be useful to see error bars for the 8 animals. Also – this can be quantified. What is the (spearman) correlation between the different networks across cells, across animals?
12.	Figure 3A caption. Ground truth is gray, not yellow.
13.	Figure 3A. The match between gridness distribution for an untrained network is puzzling. Is this due to initialization of the states? Is this due to some other property? Tuning of hyperparameters?
14.	Figure 3A. The distribution of gridness in the Dordek et al paper is between 0.8 and 1.2. How come it is so different here?
15.	L212 “were is similar rank order”. Please quantify across animals.
16.	L217 “also also”
17.	L219 the -> they
18.	L223 grid score is explained here, but used in figure 1.
19.	L238 “KS=0.1 significance level”. Why is this level special?
20.	L238 “KS-distance was generally in line” – please quantify
21.	Fig 4A – It is very problematic to draw a regression line and state that “the empirical relationship is comparatively strong”
22.	L244. “which” typo
23.	L255. The motivation for this comparison is unclear.
24.	L306: “it is quite reasonable”. Perhaps. Perhaps not.
25.	The discussion is extremely short. Why do untrained networks have a “good” gridness distribution? Why are heterogeneous cells more important than grid cells? How does this relate to place cell literature? Mixed selectivity in general?


**Time Spent Reviewing:**

4

---

> ### Author Response · Authors · 2021-08-10
> **Response to Reviewer PoER (Part 1 of 3)**
>
> - The main tool used is consistency... I found very little explanation of the motivation for this choice of metric. The authors ask … “What firing patterns of MEC cell populations are common across multiple animals?” This is a great question, but [the authors don’t answer it]. It could be that one animal has grid cells and another animal has stripe cells, but they can be linearly combined to give one another.
>
> *Our core approach is the idea that  “the artificial neurons of models should be judged to be similar to the real neurons of animals using the strictest mapping class by which the neural populations in one animal are transformable to those of any other”.  Thus, the question becomes, what is the strictest transform class such that the neural responses of one mouse’s MEC resemble those of another mouse?  This provides a notion of what properties of firing patterns are reproducible across animals, and are therefore good targets for explaining using models.
> The strictest place to start would be to try to find, for each neuron in a “target animal”, an exactly equivalent neuron in any other animal.  This is the “1-1 mapping” we evaluate in the paper. If it had worked well, we would have eschewed linear transforms, and evaluated models by forcing their individual units to map to single neurons in the data.   In essence, if any individual neuron in the brain of one animal has a corresponding neuron in all other animals, just animal to animal reproducibility then sets a high bar for a modelling: good models of this brain region should also have individual neurons that map to individual neurons in the brain, in terms of their firing properties.
> However, in practice we find that 1-1 mapping does *not* actually produce a good mapping between real animals’ populations.   In other words, it is too “strict”.  Since individual neurons in one animal are not reliably found in all other animals, matching models to brains at the level of individual neurons is not a good target for modelling; since such reproducibility does not exist across animals, we should not demand it of our models. Instead, the validity of a model should be judged at a coarser level, using a less restrictive class of similarity transformations - for example by falling back to the choice of (some subset of)  linear mappings, and using regularization as a way to control how strict the set of linear mappings actually is.  What we find is that with the properly strict regularization type (ridge), the neural population in one animal can be seen to be similar to those in another.
> Your hypothetical scenario of two mice with apparently non-overlapping cell types is very interesting because it actually illuminates the subtleties of this process.  There are two basic ways to interpret your hypothetical, though:*
>
> 1. *In one version, we have information that all the animals in the population are divided into these two types: animals with only stripe cells and animals with only grid cells -- i.e. the population is made up of “Strippies” and “Griddies”.  In this case our methodology would say: ok, despite the apparent divergence between Strippies and Griddies, we have to find a transform class that causes Strippies to be inter-transformable either to other Strippies or just Griddies, and vice versa, but would not allow mixtures of Strippies and Griddies to arise.   A subset of Linear transforms might be powerful enough to achieve that, but it would have to be a very special subset, so as not to allow for transforms to mixed strip-grid states -- which by supposition don’t exist in the population.  (And in mathematical terms, this subset would almost certainly not be a “subspace” of the space of linear transforms, because linear combinations of strippy-to-griddy and griddy-to-stripped transforms would almost certainly not be allowable.)
> Now let’s take the next step of comparing animals to models. Suppose we could find a “good model” for this context -- one that under this *same* class of transforms would be able to predict neuronal responses in both Strippies or Griddies accurately.   However -- and here’s the key point -- any candidate such model would itself *have* to look like either a Strippy or a Griddy -- since the transform class being used is, by definition, only able to transform strippies to griddies, vice versa, but nothing else.  Ultimately, we’d end up with a population of multiple models, some of which were Griddies and some of which were Strippies.  Analyzing the similarities and differences in the structure of these models would then be a very interesting next step. In other words, our method should, at least in theory, detect the interesting divergent structure in the population and handle it accordingly.*
>
> 2. *In another version of the hypothetical, the two “all stripes” and “all grids” animals are the two extremes of continuum in which animals in the population can empirically have *any* proportion of stripes and grid cells -- and likely lots of other mixture cell types -- in their MECs. In this case, the class of transforms between animals would be best chosen as linear transforms with unconstrained sparsity -- and if regularization is useful at all, regularization constants would have to be widely distributed.  The class of models that would work here would itself contain versions with widely varying fractions of stripes and grids.  In this case, the apparently striking difference between stripes vs grid cells would probably be best interpreted as being basically superficial -- something that at first glance might look important, but probably is just epiphenomenal to functionality.*
>
> 3. *The situation we actually observe in practice is not like either of these extreme hypotheticals, though. Real neurons in mice are not segregated like this, either within an animal or between animals in the population.  Most animals seem to have similar mixtures of cell types, though with potentially somewhat small differences in the proportion of  each.  To illustrate this, we have produced plots of the distribution of grid-scores of cells in our population on a per-animal basis -- we will add these as a supplementary figure in the revised version of the paper.  These distributions are all unimodal distributions, which while somewhat variable from each other, are fairly overlapping.  Thus, unlike the hypothetical Strippy-vs-Griddy case, we see no evidence for a strong restriction on the class of linear transforms, except for that imposed by regularization itself -- and this restriction ends up forcing fairly sparse mappings between individuals, largely preserving the prevalences of grid cells.  This is also consistent with our results on grid-score distribution, in which we find that the models that work well under regression metric are also those that match the grid-score distribution best, even without any linear remapping occurring.*
>
> *What’s the upshot of these observations?  Our interpretation is that the answer to the question of “What firing patterns are common across animals?” is: “The firing patterns that are common across animals are the aspects of the patterns as observed in any one animal that are left invariant by ridge-regularized linear transform”.  This isn’t a word model of what the commonalities “look like” in a simple way, but it is a quantitatively well-formulated and actionable characterization that is suitable for asking whether these same commonalities are present in a computational model.*
>
> *In essence, this highlights our philosophy: find the simplest transform class that can reliably transform firing patterns in any one animal to firing patterns in another animal, and then the equivalence classes of such firing patterns become “common across animals” and models can be judged by how well they can generate the same equivalence classes of firing patterns using the same transform class as animals can of each other.  We believe this is a principled way to compare high dimensional neural representations between models and brains, with a principled ceiling of high performance obtained by comparing brains to brains.*

---

> > ### Author Response · Authors · 2021-08-10
> > **Response to Reviewer PoER (Part 2 of 3)**
> >
> > - The authors chose a linear mapping from a population of neurons into a single neuron. Crucially, the train and test data are rate maps and the test data are 50% of the bins, under a random choice… This means that the test data is not that different from the train data. ...
> >
> > *This is a really key question. The issue of generalization is something we ourselves have worked on over the past months.  We’ve performed a series of experiments using other approaches to train/test generalization, including:*
> >
> > 1. *The random 50% splitting that we already showed in the paper.*
> >
> > 2. *Reduced random splittings in which many fewer than 50% of the bins are used in training, and all the rest are used for testing.   This approach is sensible because it turns out that even with 10% or fewer of the spatial bins in training, inter-animal consistency scores remain high.*
> >
> > 3. *Spatially-localized train-test splits, e.g. where the regression training is done on the left or right half of the spatial bins, and then test is done on other half, or similarly where the training is done on one (or three) quadrants and the test is performed on the remaining three (or one) quadrant(s).*
> >
> > 4. *Fourier-based train-test splitting, where we train on data from localized subsets of the autocorrelogram and test on the remainder.  This method is good at picking up periodic structure in cells.*
> >
> > 5. *Representational Similarity Analysis (RSA).  Instead of doing regression, here we compute the cross-neuron conditionXcondition correlation matrix for each population (e.g two animals or an animal and the model), and then look at the (spearman rank) correlation of these matrices between the two populations.  This is an approach, popularized by Kriegeskorte et al. 2008, has been used in the human fMRI literature extensively. The “pro” of this approach is that it avoids training/test entirely.   But the “con” is that it actually empirically ends up setting the inter-animal consistency ceiling quite low, compressing the dynamic range of our ability to separate models.   This is because actually the animals have different prevalences of unit types, while the RSA approach is very “prevalence-sensitive” -- and so ends up treating animals as if they were very different, thus lowering the noise ceiling of inter-animal consistency, and correspondingly making it rather easy for not-so-good models to look like they’re ok.*
> >
> > 6. *“Minimal characteristic binning” in which, for each target cell in MEC, we seek to find the *minimal* set of spatial locations needed to map it effectively from other MEC cells, e.g. a small number of (target-specific) bins such that matching source neurons to the target just on those bins retains high predictivity on all remaining bins.  Then we’d ask how good any model is at matching each target, using the *same* minimal set of bins selected for that target cell.  This procedure seems ideal, since it is not biased toward (e.g.) periodic cells, but can leverage such structure when present (e.g. for grid cells, the peak and trough of two successive grid locations would suffice as “minimal characteristic bins”).  The only real “con” of this procedure is that it is computationally very expensive.*
> >
> > *We’ve tried approaches 1-5 and are in the process of trying approach 6.* **What we’ve found is essentially all the results of the paper remain essentially the same independently of which approach we use.** *Specifically: heterogeneous cells remain just as reliable as grid cells no matter what splitting approach is used, the rank order of the models is largely not changed, and the fraction of explained variance (e.g. the model-to-data match relative to the noise ceiling of the inter-animal reliability) remains very similar -- so that the best models still explain nearly all the variability. These results show that the key ideas of the paper are quite robust to the concern the reviewer mentions, and will include figures showing this in the supplement of the revision.*
> >
> > - It would strengthen the conclusions if the authors check a range of [grid] thresholds....
> >
> > *For results in Fig.1D,  Fig.1E,  Fig.4C, and Fig 4D, we have performed robustness checks, comparing non-grid cells (threshold < 0.3) to populations of with minimal grid score t_min=0.3, 0.5, 0.7, 0.9, and 1.1.  We’ve found that none of our results depend sensitively on grid-score threshold, and will include new figures quantifying this in the revision.*
> >
> > - In general, using different trajectories, and actual responses of the neurons (instead of rate maps) might be a more fair comparison.
> >
> > *Agreed. The reason we didn’t do this here is that our data don’t have a lot of repeated samples of the same trajectories across different animals that would allow us to assess inter-animal consistency at that fine grain.   Working with rate maps seemed like a good place to start, but going forward, we plan to try to account for responses on actual trajectories.   Possibilities for how to do this include:*
> >
> > 1. *Collect quite a bit more data so we have a dense map with multiple trials in each position X head-direction X running-speed bin, and then essentially perform the same analysis we have here already done.  This is a straightforward approach but is just very experimentally intensive.*
> >
> > 2. *Marginalize over some of the axes, e.g. separately look at the head-direction and running-speed bins, averaged over position.  This is easy to do, but eliminates a lot of the possibility for interactions between the variables.*
> >
> > 3. *Forget about estimating inter-animal consistency and just go ahead and compare trajectories, first fitting models to animals on some of the trajectory and then comparing outcomes on later portions of the trajectory (for example).  The danger here is that if models don’t fit well to this data in an absolute sense, it will be hard to tell if that’s because the models are bad or because actually the responses are highly variable.*
> >
> > *We will probably end up trying all three methods and checking whether we get consistent answers across them.*
> >
> > - L180: models are substantially different. How consistent is the pattern of figure 2 across different animals?
> >
> > *Good question! We have broken out the results of Fig 2A on a per-animal basis, and for each pair of animals computed the spearman rank correlation for the vector of predictivities across models. Across animal pairs, the minimal spearman correlation is 0.94, and the median is 0.98 -- eg. all the animals have essentially the same model preference ranking.  The matrix of these pairwise correlations will be included in the revised paper as a supplementary figure.*
> >
> > - Figure 3A. The match between gridness distribution for an untrained network is puzzling. Is this due to initialization of the states? Is this due to some other property? Tuning of hyperparameters?
> >
> > *Really interesting question, but one we don’t fully know the answer to.  We didn’t do any hyperparameter tuning in our work, so we suspect that the answer is that grid cells are essentially the result of performing a kind of dimension reduction on place cells, and some architectures might essentially mimic this outcome even with random filters.  In understanding this, it is important to stress that networks with “random filters” are NOT equivalent to random functions [Saxe & Ng 2011].  As has been seen repeatedly with neural networks, with the “right architecture”, the functions computed by the network even with random filters are often (a) pretty useful for tasks and (b) a good deal more similar to real neurons than untrained or trained neurons from networks with the “wrong architecture” [Yamins et al. 2013, 2014].*
> >
> > *A well-explored example of this is from vision, where it’s found that for convnets with “good” architectures (i.e. that end up solving visual tasks), the **untrained** states are much better predictors of neural responses than “bad architectures” [Yamins et al. 2014].  The model-neuron match numbers for such models does also increase after task training as well, so about 50% of the fit is due to architecture alone and about 50% due to filter training.   A similar outcome has been observed in auditory cortex [Kell et al. 2018].   We suspect something similar is going on here, as also suggested by the improved neural regression fit here in the MEC case for untrained versions of the better architectures relative to the worse architectures.  This is a good issue for future work.*
> >
> > - Figure 3A. The distribution of gridness in the Dordek et al paper is between 0.8 and 1.2. How come it is so different here?
> >
> > *Dordek et. al. uses a different metric. The metric we use [Langston et al. 2010, Butler et al. 2019] is the mean correlation at 60 degrees and 120 degrees minus the mean correlation at 30 degrees, 90 degrees, and 150 degrees (see sec A4 of Supplement for more info). The gridness reported in Dordek et al. is with respect to 60 and 90 degrees alone.   Our choice of metric maintains consistency with the original data papers on which our analyses are based [Mallory et al. 2021, Butler et al. 2019].*
> >
> > - L212 “were of similar rank order”. Please quantify across animals.
> >
> > *The spearman rank order correlation between the equivalent bars in Fig 2A and 2B is 0.74 (pval<<0.01).  Moreover, the worst and best models are preserved between the two rank orderings.*
> >
> > - L238 “KS=0.1 significance level”. Why is this level special?
> >
> > *No particular specialness, it’s just reported for concreteness :)  We also report the raw KS statistics (see Fig 3).*
> >
> > - L238 “KS-distance was generally in line” – please quantify
> >
> > *This is quantified in supplementary figure S2.*

---

> > > ### Author Response · Authors · 2021-08-10
> > > **Response to Reviewer PoER (Part 3 of 3)**
> > >
> > > - L255. The motivation for this comparison is unclear.
> > >
> > > *To recap, the question being addressed in this section (section 4) is: what is the functional role of heterogeneous cells?  One way to address this question is to ask: do task-trained models that are better at a given task than (say) classical models differentially improve on explaining heterogeneous cells.  To the extent this is true, it is evidence (albeit weak) that the role of the heterogeneous cells (at least, in the model) is to help perform the task for which the network is trained.  This is what we find.  Of course, this is not the only test that one would want to address this topic, which is why in this section as a whole we show three distinct tests that try to get at this larger issue from several points of view.*
> > >
> > > - L306: “it is quite reasonable”. Perhaps. Perhaps not.
> > >
> > > *Fair enough.  But in a way, the results that we do show in the section (even though we don’t explicitly implement an end-to-end RL model) support the reasonable-ness of this idea.  What we’ve done is to “act out” the results of a successful RL model, where the animal goes to the reward faster and stays there longer than otherwise -- while still having the model just do usual place-cell construction as its explicit task.  The fact that these models seem to work reasonably well to capture reward-modulated responses suggests it’s not a big leap that an end-to-end model that achieved something similar might also match neural responses.   Of course, given our earlier results that pure position estimation is not as good as explicit place-cell construction at creating MEC-like representations, the interesting question is whether an end-to-end RL model would explicitly construct place cells if it weren’t explicitly forward to do so…. An interesting question for future work!*

---

> > > > ### Comment · Reviewer_PoER · 2021-08-22
> > > > **A comment and further questions**
> > > >
> > > > I thank the authors for their detailed explanations of my questions, and those of the other reviewers. I now understand the results of the paper much better.
> > > >
> > > > I want to raise a point that is related to the level of details in the authors' responses - clarity.
> > > >
> > > > There are many details that were either missing or not explained well in the original submission. The need for such detailed explanations is an indirect testimony to this issue. While the clarity of the authors' responses convinces me that the authors can clearly explain their motivation and methods, I can only rely on hope and optimism that this clarity will be fully integrated into a revised manuscript.
> > > >
> > > > A few concrete questions:
> > > > * Figure 1C & 1D & S3. What is the exact definition of *number of source cells per target unit*? If the predictivity is different for each target, what was the threshold used to determine the needed number of source units? Was this done by choosing the top-correlated cells? Was this done by a threshold on the parameters following ridge regression? Something else?
> > > > * What does the vertical dashed line in figure 1C represent? There is an unfortunate visual illusion of dividing the red and blue labels of heterogeneous and grid cells relative to this line.
> > > > * The controls of figure 2A (Velocity, linear place cell): It was quite hard to find the definition of linear place cell control. I could not find the definition of *velocity input control*.
> > > > * Fraction of source population: do we have this information for the models?
> > > > * Correlations using 6 models (Figure S2 for instance). I think it’s hard to make these claims based on such small numbers. Perhaps running 100 instances of each model is a more suited tool for these correlation questions.

---

> > > > > ### Author Response · Authors · 2021-08-24
> > > > > **Response to Reviewer PoER**
> > > > >
> > > > > - I want to raise a point that is related to the level of details in the authors' responses - clarity. There are many details that were either missing or not explained well in the original submission. The need for such detailed explanations is an indirect testimony to this issue. While the clarity of the authors' responses convinces me that the authors can clearly explain their motivation and methods, I can only rely on hope and optimism that this clarity will be fully integrated into a revised manuscript.
> > > > >
> > > > > *We fully intend to integrate what we have learned from the review process into the revised manuscript.  In fact, we are excited to do this, because we feel that the clarifications that responding to the reviews have prompted are very interesting in their own right.  The extra page that is afforded to accepted papers will give us the opportunity to discuss many of the issues raised in greater detail, and we very much look forward to the opportunity to do so.*
> > > > >
> > > > > *We now turn to answering the specific new questions you’ve asked:*
> > > > >
> > > > > - Figure 1C & 1D & S3. What is the exact definition of number of source cells per target unit? If the predictivity is different for each target, what was the threshold used to determine the needed number of source units? Was this done by choosing the top-correlated cells? Was this done by a threshold on the parameters following ridge regression? Something else?
> > > > >
> > > > > *The analysis in this part of the paper was intended to evaluate the intrinsic “complexity” of heterogeneous cells in the ground truth data relative to grid cells, ultimately as a target of explanation for the models. The “fraction of source population” is defined in Supplementary Section A3, lines 494-496, and refers to the top most correlated (via Pearson correlation on the training position bins) percentage of units in the source population for each target unit. In Figure 1D, we then select the minimum fraction of source population units for each target unit needed by the mapping transform to reach that target unit’s inter-animal consistency, as explained in the Figure 1 caption. We will additionally include this description in the revised version of the main text to ensure clarity.*
> > > > >
> > > > > - What does the vertical dashed line in figure 1C represent? There is an unfortunate visual illusion of dividing the red and blue labels of heterogeneous and grid cells relative to this line.
> > > > >
> > > > > *The vertical dashed line demarcates the grid score threshold we use (0.3) for demarcating whether a cell is a grid cell or heterogeneous cell, which is precisely why the red labels on the right side of this line demarcate grid-like units, and the blue labels to the left side of this line demarcate the heterogeneous units.*
> > > > >
> > > > > - The controls of figure 2A (Velocity, linear place cell): It was quite hard to find the definition of linear place cell control. I could not find the definition of velocity input control.
> > > > >
> > > > > *Given the page limitations on the initial submission, we could not put the definitions of each control in the main text, and therefore originally put them in supplementary material, as we had to do for the linear place cell control (defined in lines 611-612 of Supplementary Section C1). We are happy to move some of these definitions to the main text in the revised version.*
> > > > >
> > > > > *Specifically: the velocity input control is application of the mapping transform (ridge regression) to the 2D velocity input (namely the speed-head direction vector) of the path integrators, and is defined in Supplementary section C4.1 in lines 625-625 and again in line 629 when we mathematically explicate how it is used by the models in equation 15. We will make it its own short subsection in C4 prior to introducing the path integrators.*
> > > > >
> > > > > - Correlations using 6 models (Figure S2 for instance). I think it’s hard to make these claims based on such small numbers. Perhaps running 100 instances of each model is a more suited tool for these correlation questions.
> > > > >
> > > > > *We agree that the correlations in Figure S2 are across a small number of models -- although this is simply meant to illustrate a general trend between the two neural metrics of comparison (predictivity under ridge regression & grid score distribution match) and is not essential to any of the main arguments of the paper.  But you’re right that having more models here to fill out the axes and see if a strong correlation emerges would be a great idea.  The question is how to do so.  One potential way to generate more models would be to create 100s of instances of models differing only in (e.g.) the random initial condition of weights.  We will be happy to try this.   The only issue that would likely arise here is that we know from our anecdotal experience that generating variability by modifying initial weights does not change the key observables here (e.g. task performance, neural match scores, &c) appreciably.  In other words, the metrics we compute are pretty stable to this type of change.  (That is a result we should include in and of itself in the revision.)  But the implication of this is that we will probably need a stronger source of variability to fill out this plot more.  To this end, we instead plan in future work to train a larger class of recurrent agent architectures (e.g. GRUs, IntersectionRNNs, UGRNNs [Collins et al., 2017]) along with varying their internal nonlinearities and loss functions, to explore more variance across the space of possible circuit structures.*

---

> > > > > > ### Comment · Reviewer_PoER · 2021-08-28
> > > > > > **Vertical line in figure 1C**
> > > > > >
> > > > > > I might be missing something, but did you answer regarding figure 1C or figure 1D/E?
> > > > > > All three panels have vertical lines.
> > > > > > All three panels have something blue on the left and something red on the right.
> > > > > > The X axis is different in figure 1C, and this panel also contains two curves.
> > > > > >
> > > > > > Can you please clarify?

---

> > > > > > > ### Author Response · Authors · 2021-08-29
> > > > > > > **Re: Vertical line in figure 1C**
> > > > > > >
> > > > > > > The vertical line in Figure 1C is just to demarcate the fraction of the source population needed for heterogeneous and grid populations to reach 0.978 correlation, which is the median inter-animal consistency across the entire neural population (denoted by the horizontal line in that panel).
> > > > > > >
> > > > > > > In Figures 1D/E, the vertical line in both panels (which each have an x-axis label of "Grid Score") demarcates the grid score we use (0.3) for demarcating whether a cell is a grid cell or heterogeneous cell.

---

### Official Review · Reviewer_vfhu · 2021-07-15

**Rating:** 8
**Confidence:** 3

**Summary:**

By adopting 'similarity transform' methods, this study compares the predictability of different computational models of MEC neurons. The best model not only can explain the firing patterns of grid cells but also that of heterogeneous cells. These findings further allow them to examine the role of heterogeneous cells which has not been clearly identified with experimental approaches. This study suggests that the heterogeneous cells might play critical roles in path integration even more than what grid cells contribute. This manuscript is well-written, and the results provide a theoretical foundation for future studies using a more experimental approach. I only have minor clarification questions and suggestions to embrace a broader readership such as neuroscientists.

**Limitations And Societal Impact:**

Yes.

**Main Review:**

1. What does it mean by that the place cell activity is predicted by the network trained in a different size arena at the MEC level? Are neurons (units) identified as grid-cells in one environment more likely to be grid-cells in the larger environments as if expanding the toroidal attractor manifold? If it is the case, how does the level of gridness / heterogeneity changes from one to the other environment? This may be hard to test in experiments since velocity inputs are different even in the same environments but it might be able to test within the computational models.

2. Can the model also predict that the reward attracts the firing fields of the grid cells, as the original data have shown? Is there any possibility that differences in the rate maps between reward+ and reward- conditions stem from different profiles of velocity (such as the animals approaching the rewards rapidly and staying longer) not by the reward itself? I see that the authors argued this a bit, but can your results suggest that this reward-associated predictability is not fully caused by the velocity profiles only but by using the reward input layer?

3. How can we interpret the finding that the place cell activity is better predicted by path integration than the actual coordinate? Is it because neighboring place cells can encode two locations that are actually far from each other? Meanwhile, the high inter-animal consistency might suggest that the place cells mapping might not be totally random but reflect some spatial properties. It would be very helpful if the authors include what the results suggest and how the readers can interpret these findings.

4. Some previous evidence suggests that place cells are independently formed from the grid cells, and often they are stabilized earlier than the grid cells (while many others support the idea of the contribution of grid cells on the place cells). Is this finding compatible with those previous studies (e.g. those studies did not take into account the effects of heterogeneous cells)? Does this study suggest that the place cells are fully dependent on and driven by the MEC neurons? If not, some limitations may need to be included.

**Time Spent Reviewing:**

5

---

> ### Author Response · Authors · 2021-08-10
> **Response to Reviewer vfhu**
>
> - What does it mean by that the place cell activity is predicted by the network trained in a different size arena at the MEC level?
>
> *The best way to understand our generalization test is that the “animal” (in this case, the model) is trained on an environment of one size; but we can then ask whether its internal MEC-like representation is actually functional task-wise and neurally predictive on other arena sizes.  This is a little bit like what we think happens with the mice in general: they’re “trained” (in some natural self-supervising way) on whatever life experience they have, and then brought to the experimental environment, where they still seem to be able to navigate fine, and have place-cell and grid-cell representations emerge.  Our generalization test is trying to capture this in some way.*
>
> - Are neurons (units) identified as grid-cells in one environment more likely to be grid-cells in the larger environments as if expanding the toroidal attractor manifold? If it is the case, how does the level of gridness / heterogeneity changes from one to the other environment?
>
> *This is a good question.  Before your question, we had known anecdotally models that generalize well overall do seem to do so by “preserving grid-ness” -- e.g. at least some cells that  have high grid-score in one arena also have high grid-score in other larger or smaller arenas.  (This is consistent with results shown by Banino et al. 2018.)   In response to this question, we’ve done an analysis to quantify this, for two models, one which generalizes well across arena sizes (LSTM-ReLu-place-cell) and one which doesn’t (LSTM-ReLu-position).  We find that for the model that generalizes, many (~50%) of the cells that look like grid cells in the small arena also maintain grid-ness in the big arena, especially the most grid-like cells.  In contrast, for the model that does *not* generalize well, very few of the grid cells as identified in the small arena (1%) are actually grid-cells in the larger arena.  This suggests that preserving some grid-like-ness (and thus its flipside, “heterogeneousness”) in a generalizable way is a contributor to model effectiveness.  [Interestingly: our experimental collaborators tell us that while this question very likely hasn’t been published on, in their own unpublished work, they find something like a 50/50 split of neurons that look like grid cells in small arenas that actually generalize to being grid cells also in substantially larger arenas.]*
>
> - Can the model also predict that the reward attracts the firing fields of the grid cells, as the original data have shown?
>
> *Good question.  Since (almost by definition), achieving essentially 100% of the neural variance explained means that any phenomena that occur in the original data must occur in the modelled data as well, so the model (with the linear mapping transform) produces this phenomena.  Since this type of phenomena is a linear invariant,  we also find that this phenomena is also present in the raw model *before* the linear mapping transform.  In short, the answer to your question is “yes”.*
>
> - Is there any possibility that differences in the rate maps between reward+ and reward- conditions stem from different profiles of velocity (such as the animals approaching the rewards rapidly and staying longer) not by the reward itself? I see that the authors argued this a bit, but can your results suggest that this reward-associated predictability is not fully caused by the velocity profiles only but by using the reward input layer?
>
> *This is an interesting question!   Actually, now that you ask about it, we think that the original data collectors’ explanation is probably more likely to be the case.  We think this is the case for two reasons.  First, the models that work well (the pink-colored high-performing bars in Fig 5b) do so *without* actually having any explicit reward input.  The only thing that’s different about them from the original models is that they have different velocity profiles:  namely, they go to the reward zone more efficiently and spend more time there.  So actually the very fact that those differential profiles are enough to cause the model to distort its grid cells in the observed way is, to our mind, suggestive that those distortions are actually just caused by velocity changes and not reward as such.   The other thing we find is that a model that *does* have direct explicit access to reward input, but doesn’t change its velocity profile in response to the reward in any meaningful way, is not that effectively predictive of neural responses, though it’s slightly better than not having reward access at all. (This can be seen in the light-purple bar in fig 5b.)  So taken together, we think these observations are some evidence in favor of the original authors’ hypothesis.  We will note this point in the revised version -- thanks for asking about it.*
>
> - How can we interpret the finding that the place cell activity is better predicted by path integration than the actual coordinate? ...
>
> *Great question!  First, we just want to be clear about what we actually found.  We didn’t actually test any models against real place cells in mouse hippocampus (HPC) -- all the data we compared models to was from MEC, not HPC.   With this in mind, what we found that is MEC-cell activity is better predicted by the model’s intermediate layer when the model’s downstream layer is optimized to replicate place cells explicitly, rather than when that downstream layer bypasses an explicit place-cell like representation and goes all the way to positional estimation.   In other words, “the model predicts MEC better when it has an explicit hippocampal-like layer rather than when it doesn’t”.*
>
> *Now, why would this be?  There are at least two interesting possibilities:*
>
> 1. *One possibility is that the problem with the direct-to-position model is that it’s not architecturally correct.  Because it only has one hidden layer, it doesn’t even have the possibility of having a separate hippocampus.   If we trained a model with an additional hidden layer, but having direct position estimation still come out the end, there’s the possibility that the earlier of the two hidden layers would be MEC-like (with many grid cells, and a bunch of heterogeneous other cells) and the later one would be HPC-like (with many place cells, and another bunch of heterogeneous other cells of a different kind). If we found that, it would basically say, “grid cells and place cells organically arise from a multi-layer architecture solving end-to-end path integration”.*
>
> 2. *Another possibility is that even with two layers, the model won’t correctly generate place or grid cells with just the end-to-end positional estimation task as the optimization target.  This result would be interpreted as saying that direct end-to-end positional estimation is just not quite the right constraint on what shapes mouse MEC/HPC -- and suggest that we search for some *other* ecological  task that forces the existence of place-like cells in the penultimate layer would have to be found.*
>
> *Either way the answer is super-interesting. This is a great question for future work we hope to do.*
>
> - Some previous evidence suggests that place cells are independently formed from the grid cells, and often they are stabilized earlier than the grid cells (while many others support the idea of the contribution of grid cells on the place cells). Is this finding compatible with those previous studies (e.g. those studies did not take into account the effects of heterogeneous cells)? Does this study suggest that the place cells are fully dependent on and driven by the MEC neurons? If not, some limitations may need to be included.
>
> *This is a good question.  Our result that having place-like cells as an explicit target seems to be needed to get the MEC population to form correctly might possibly be interpreted as support for the idea that place cells should be stabilized earlier than grid cells.  However, the stronger evidence would come from the kinds of explorations we suggested above in response to your previous question.  Following that train of thought further:*
>
> 1. *If having a two-hidden-layer architecture with a single end-to-end loss (position estimation or otherwise) as the loss function worked well to get both MEC and HPC populations (for both stereotypical and heterogeneous) cells, that would suggest that the right interpretation is that it’s not so much that the place cells are forming independently, but rather that both populations are part of a larger whole that needs both the correct inputs upstream and the correct target downstream to be correct to see either.  Of course, even if this did work, one might still see place cells in the HPC-like layer stabilize earlier than grid cells in the MEC-like layer, just because the dynamics of learning might cause a more down-stream layer to stabilize earlier -- and if that were true, it would suggest that the previous evidence is more an “epiphenomenon”  than a deep result of causality flowing from place cells to grid cells.*
> 2. *On the other hand, if we actually needed to have a separate loss function imposed in the middle of the system to generated the place cells as such (and thus generate the upstream grid-cells), but we a separate downstream loss function to generate the rest of the (non-place cell) HPC population, this suggests that the Place-Cells-Then-MEC-cells causality theory might be more a correct way to describe things.*
>
> *Frankly our bet here is on the first of these possibilities because it seems simpler, but either way, again, very interesting.*

---

> > ### Comment · Reviewer_vfhu · 2021-08-28
> > **Thank you for your detailed explanations to my questions**
> >
> > Thank you for giving a more detailed explanation which helps me to understand your work better.
> > I'm also looking forward to seeing the future works that you mentioned.
> >
> > I realize some of my questions could have been clearer.
> > I would appreciate it if the authors can make some additional comments on the following.
> >
> > 1. My first question can be put in this way. Can the finding that the MEG activity trained from one arena predicting that in a larger arena be driven by that the same toroidal attractor model is applied to the two open spaces arena in different sizes? If it is then, the predictability can be decreased if the testing arena has different environmental constraints from the training arena, such as the square one to the trapezoid arena?
> >
> > 2. In the reward trials compared to the non-reward trials, the **variation** of the grid-cells and heterogenous cells might not be different from each other while the firing fields in the grid cells might change to afford efficient codings of the route to the rewarding location. Since the authors mentioned that *"this phenomena is also present in the raw model before the linear mapping transform"*, it seems that the changes in the firing rate of the grid cells of the same animal from the non-rewarding trials are indeed predicted by the model when the rewards are introduced. However, this cannot be shown with the variance comparisons. I would appreciate much if I can find the very results mentioned in the response behind *"In short, the answer to your question is “yes”."* in the supplementary results.

---

> > > ### Author Response · Authors · 2021-08-31
> > > **Re: Reviewer vfhu Follow Up Questions**
> > >
> > > - My first question can be put in this way. Can the finding that the MEG activity trained from one arena predicting that in a larger arena be driven by that the same toroidal attractor model is applied to the two open spaces arena in different sizes? If it is then, the predictability can be decreased if the testing arena has different environmental constraints from the training arena, such as the square one to the trapezoid arena?
> > >
> > > *Ah, ok -- yes, we better understand the question now, thanks for the clarification. There are really three sub-questions you’re asking:*
> > >
> > > 1. *Are the grid cells of the learned models that match the existing neural data well (e.g. like the LSTM ReLU model) best described mathematically through the embedding of some kind of toroidal attractor, and if so, what does that imply for how cell properties change as environment constraints change?*
> > >
> > > 2. *Regardless of the answer to the first question, given that we’ve only reported results on square arenas, what can we say about  what happens to the grid cells of the models in non-square arenas?*
> > >
> > > 3. *And, regardless of the answer to question 2, how well does the model do in matching real neural data in non-square arenas -- either for grid cells or (at least as importantly) for heterogeneous cells?*
> > >
> > > *All three of these questions are interesting, and of course the answer to the first one was affirmative, it would constrain the answer to the second one -- but even if not, we can still think about the second one empirically.*
> > >
> > > *Ok, so now to actually (attempt to) answer these questions:*
> > >
> > > 1. *To some extent, the experimental and mathematical analysis of [Sorscher et. al. 2020] suggests your idea may be right -- namely, that the grid-cells of the learned model can be seen as an approximation of toroidal dynamics.  However, the mathematical results there haven’t been extended yet to directly answer your question of how response properties of the (learned) model should change as a function of arena shape.   It is possible to imagine adapting their techniques to answer this -- e.g. something like doing their eigenmode analysis in the non-square domain -- but we haven’t attempted to do anything like that yet.  It’s a good question for future work!*
> > >
> > > 2. *We actually haven’t tried looking at responses on non-square arenas yet.   If you look at [Cueva and Wei, 2018] you’ll see that they did something like that for their models, but those are not the best performing models overall.  So it would be worth doing again with the current best models.  We haven’t done it yet just because it would take us some time to retool our code to handle non-square arenas, and we know there’s some time pressure to respond given the review cycle.  However, we will start working on this now and if we get an answer before it’s too late, we will post it.  (Regardless, we will put what we find into the supplement of the paper.)*
> > >
> > > 3. *It is definitely possible that the models that look right in square arenas don’t generalize to non-square arenas -- meaning that they might not match neural data there.   One thing we can say is that models trained in 2D do seem to generalize to match neural data in 1D -- that’s shown by the results in our Fig 2B of the paper.  That’s evidence that the models generalize reasonably well in some contexts -- and suggests (albeit weakly) that models would generalize to non-square 2D arenas. Unfortunately we ourselves don’t yet have quantitative experimental data in non-square arenas so we can’t test these hypotheses immediately.  However, there are some experimental reports about non-square arenas from the literature -- with rather interesting results, see e.g. [Krupic, Bauza, Burton, & O’Keefe, Science 2018].  One medium-term direction that we could pursue is to see whether models behave in non-square arenas the way that is report in those papers -- though some substantial work would have to be done to formalize the effects they report in model-computable metrics (we tried to do this quickly in response to your question, but realized it was not something we could easily do in a few days). But more profoundly, your question is really the tip of the iceberg. If we had more data, we’d actually want to push the models really hard -- not just to test their generalization to non-square open areas (as done in e.g. Krupic et al), but also to environments that depart considerably from open arenas (e.g. with walls, enclosures, rich cues, complex reward patterns, and so forth).   We are not confident that the current best models would necessarily perfectly match the neural data in that much richer type of environment, and think of this kind of hard test as exactly the next step in experimental work that should be done to falsify our models.  In fact, it would actually be good in our view if the models that currently appeared to work in the open arenas were meaningfully **falsified** in much more complex environments.  We don’t think LSTM ReLU is the ultimate answer to MEC, and it would be great to show that the harder constraints of a richer dataset could reject our current set of models --  and thus help refine/improve our picture of a stronger overall model that worked in **all** the situations.  (Such a model might have to be considerably more interesting than a simple LSTM ReLU… which would be very interesting.)
> > > We are trying to convince our experimental partners to try this kind of experiment -- and one of the main things we’ll discuss in the extended discussion section of the revised paper (where we get an extra page) is this very issue: that is, we think our results here make the strong recommendation that the new experiments be done to test generalization in much richer environments, and that is really time to push these experiments far past simple open arenas. **We will accordingly discuss the potential limitations of our current model very explicitly in this discussion.***
> > >
> > > *Apologies -- we do realize that we haven’t been able to provide final answers to any of your current questions -- they’re hard but very interesting questions that (fortunately or unfortunately) mostly belong to future work.  But we will try to post the answer to what happens to the models in non-square arenas as soon as we’re able to.*
> > >
> > > - In the reward trials compared to the non-reward trials, the **variation** of the grid-cells and heterogenous cells might not be different from each other while the firing fields in the grid cells might change to afford efficient codings of the route to the rewarding location. Since the authors mentioned that "this phenomena is also present in the raw model before the linear mapping transform", it seems that the changes in the firing rate of the grid cells of the same animal from the non-rewarding trials are indeed predicted by the model when the rewards are introduced. However, this cannot be shown with the variance comparisons. I would appreciate much if I can find the very results mentioned in the response behind "In short, the answer to your question is “yes”." in the supplementary results.
> > >
> > > *Ah, we (think we) understand the question better now.  Basically we understand you to be asking: does the raw model, pre-linear transform, exhibit the phenomenology of having peaks of grid cells up-regulated by the presence of reward? (let us know if this isn’t what you’re asking about)*
> > >
> > > *This is something we would like to check directly in comparison to the original empirical result in the Science paper in which this phenomenon was reported.  We have the relevant data for checking this already (e.g the original source data from that paper), but we don’t yet have quite the right metric for reproducing the quantitative analysis that led to that result.  In response to your question, we did some thinking about how we would reproduce the result in the actual data (putting aside the models for a moment) but these lead us to realize there are some subtle details about how to measure the phenomenon itself correctly -- so that we can’t yet reproduce the original phenomenon yet even though we have the raw data.  Once the original authors of that paper share with us the analysis code they originally used to find the phenomenon in the first place, we will reproduce their result in the data, and assuming that works, check it out in the models.  We likely won’t be able to do this before the review period ends, but we certainly will be able to get it before publication, and we’ll put the final result (whatever it is) in the supplement of the revised paper.  Either way, the answer will be interesting!*

---

### Official Review · Reviewer_7Y6g · 2021-07-21

**Rating:** 8
**Confidence:** 3

**Summary:**

The authors employ RNN models under different conditions to model MEC cells with both well-understood (e.g., place cells) and poorly-understood (heterogeneous cells) tuning properties. By training the network to minimize the place cell-mediated path integration, the network performed incredibly well in explaining MEC-cell dynamics. By removing neurons in the model, the network's path-integration performance was found to be at least as dependent on the model's heterogeneous neurons as on the stereotypical ones. The authors then finish by introducing a reward-based foraging model, again finding responses that match experimental results very well.

**Limitations And Societal Impact:**

Yes

**Main Review:**

By testing several different neural network models with different loss functions, the authors provide a strong argument in support of heterogenous MEC cells' functionality as key to path integration, in spite of a lack of clear tuning properties. They have integrated experimental data from different animals together well, and have demonstrated that neural network models can provide a key roll in understanding biological neurons and neuronal networks.


**Time Spent Reviewing:**

3.5

---

> ### Author Response · Authors · 2021-08-10
> **Response to Reviewer 7Y6g**
>
> We thank the reviewer for their positive comments about our work!

---

### Decision · Program_Chairs · 2021-09-27

**Decision:**

Accept (Spotlight)

**Comment:**

Dear Authors,

congratulations on your paper being accepted at Neurips. Reviewers founds this a highly interesting and intriguing submission, with results that will likely be of interest to a wide range of researchers, and in particular researchers aiming to understand how diverse cells in the MEC underly the ability of animals to navigate. At the same time, this paper also triggered substantial discussion-- the primary concern by the reviewers was that this was a dense paper loaded with results, but low on detailed explanations, which made it challenging for the reviewers to assess the validity of the methods and results. During the discussion phase, extensive additional explanations and clarifications were given, and we decided that the paper -- with these additional explanations -- would be a great addition to Neurips. We urge you to ensure that these explanations also appear in the final paper (or, in most cases, in the supplement).

With best regards, your AC